# Behavior of As/As$_x$S$_y$ in Neutral and Oxidizing Atmospheres at High Temperatures—An Overview

Kristhobal Castro [1] , Eduardo Balladares [1] , Oscar Jerez [2], Manuel Pérez-Tello [3] and Álvaro Aracena [4,*]

1 Departamento de Ingeniería Metalúrgica, Facultad de Ingeniería, Universidad de Concepción, Concepcion 4070371, Chile; kriscastro@udec.cl (K.C.); eballada@udec.cl (E.B.)
2 Instituto de Geología Económica Aplicada, Universidad de Concepción, Concepcion 4070386, Chile; ojerez@udec.cl
3 Departamento de Ingeniería Química y Metalurgia, Universidad de Sonora, Hermosillo 83000, Mexico; manuel.perez@unison.mx
4 Escuela de Ingeniería Química, Pontificia Universidad Católica de Valparaíso, Avenida Brasil 2162, Valparaiso 2362854, Chile
* Correspondence: alvaro.aracena@pucv.cl

**Abstract:** The reaction mechanisms in As and As-S systems during their oxidation and/or thermal decomposition are complex to describe due to the physicochemical characteristics of arsenic and its sulfides; the information highlighted in the literature was analyzed and correlated to determinate the predominant phases and reaction mechanisms during the thermal decomposition and oxidation of arsenic, in its elemental form and in sulfurate phases. As a result of this analysis, it was determined that the predominant phases are mainly composed of allotropies of arsenic, sulfides, and dimers. In addition, reaction mechanisms are provided that describe the behavior of arsenic and its sulfides during its thermal decomposition and oxidation.

**Keywords:** reaction mechanisms; arsenic sulfide; thermal decomposition; oxidation; roasting

## 1. Introduction

The increasing presence of arsenic in copper sulfide ores presents one of the greatest challenges for the metallurgical industry in Chile, which is the world's leading producer of mine copper. Although the sharp increase in arsenic content in sulfide ores has been confirmed for a couple of decades, the definitive solution for the safe and economical removal of arsenic from these ores has not materialized yet. The great difficulty in finding the design of such a method is related to the arsenic's intrinsic characteristics and its compounds. Arsenic has multiple valence states (−3, +3, and +5), its compounds have, in many cases, high vapor pressures and activity coefficients, as well as a high chemical activity [1]; therefore, this element is present in significant concentrations in all phases within a pyrometallurgical melting-conversion process, making its separation difficult. However, the impact generated by the presence of arsenic is much broader. Non-ferrous metal smelters, particularly copper smelters, are concerned about the increase in their production costs associated with arsenic handling and, eventually, its treatment and disposal. In addition, it interferes with the process, since it is not enough to operate under the set of conditions that optimize metal recovery, but the best set of parameters and variables must also optimize its purity. This last aspect is relevant given that until 15 years ago the country's sulfide copper ores typically contained 0.5% arsenic, a value that today can reach 4% in some cases [2].

Prior to pyrometallurgical processes, the ore processing with increasing arsenic contents establishes new restrictions, as the concentrate to be produced must maintain the same low levels of contamination in order to avoid penalties imposed by the market [3–5].

On the other hand, and no less important, metallurgical plants processing ores with high arsenic contents must adjust their operation to comply with increasingly restrictive environmental regulations, which imposes additional limitations [6].

Removal and/or neutralization of arsenic from pyrometallurgical processes has been a common practice in the treatment of copper ores for decades, either as a pre-treatment or after smelting, processing the smelter dusts by aqueous media to obtain calcium and iron arsenate precipitates for final disposal. Similarly, arsenic can be captured in the form of trioxide, which is of commercial interest.

Whatever the technology used to process arsenic-containing sulfide concentrates of non-ferrous metals, a series of solid, liquid, and/or gaseous streams containing arsenic in varying amounts will be generated. Thus, it is usually required to condense and stabilize it from gases [7–12], precipitate it from leaching solutions [13,14] or wastewaters [15–17], separate it from copper in smelter slag flotation [18], and/or minimize arsenic emissions to the atmosphere [19].

The present and future scenario of pyrometallurgy poses important challenges for operators and researchers, who will have to deepen their knowledge regarding arsenic behavior in the systems that will be implemented, through the detailed characterization of the products and the study of the mechanisms involved in these processes.

The reaction mechanisms in As-S and As-S-O systems during their oxidation and/or thermal decomposition are complex to describe because their compounds sublimate or decompose with the formation of more than one phase [20] and, in addition, arsenic compounds have a low melting temperature [21,22], which causes multiple phases to volatilize in parallel. According to Landsberg et al. [23] and Ruiz et al. [24], arsenic, its sulfides and oxides have a relatively high vapor pressure, thus arsenic is a highly volatile element; it and its oxides are volatile at roasting and smelting temperatures [25,26]. Moreover, it is often considered as a semi-volatile element [27] since it is gradually eliminated when calcined.

Knowing the behavior of arsenic is important because it is one of the most common toxic impurities found in copper concentrates and, in the form of sulfides, it is the main component of almost all non-ferrous ores [28]. Because of this, its presence generates major problems from an environmental point of view [29]. In copper concentrates, arsenic can exist as a wide variety of minerals, such as orpiment ($As_2S_3$), arsenopyrite (FeAsS), enargite ($Cu_3AsS_4$), and tennantite ($Cu_{12}As_4S_{13}$) [30], among others.

As-S systems are of interest to many researchers due to the scarcity of reaction mechanisms, as well as the absence of complete databases with thermodynamic and kinetic parameters describing the behavior of these species. Furthermore, these values vary depending on the source [28,31] from which they are obtained; therefore, the behavior of arsenic cannot be predicted with the current equilibrium data found in the literature, and thus there is a gap that needs to be addressed. A bibliographic review is presented below, where the most relevant information showed in the literature on the possible reaction mechanisms during the oxidation and thermal decomposition of arsenic from its elemental form and its sulfide phases are correlated, revealing the predominant species mentioned for each system. The term thermal decomposition is applied when a species decomposes to its most stable phase only by the effect of temperature; therefore, those articles in which work was carried out under an inert atmosphere were considered.

## 2. Thermal Decomposition

During thermal decomposition, elemental arsenic volatilizes as $As_4$, $As_3$, $As_2$, and As [25,32–35]; whereas, sulfides present a large number of reactions when subjected to thermal analysis studies which involve mass and energy variations [36]. In neutral roasting, arsenic volatilizes as sulfide compounds of the form $As_4S_4$ [37], $As_2S_3$ in the form of the dimer $As_4S_6$ and AsS [25,38], which have advantage over oxides of being compounds of low solubility in water and lower toxicity [20]. According to the information analyzed [20–54], the predominant gaseous species during elemental arsenic thermal decomposition and some of its sulfides are summarized in Figure 1, which will be discussed later.

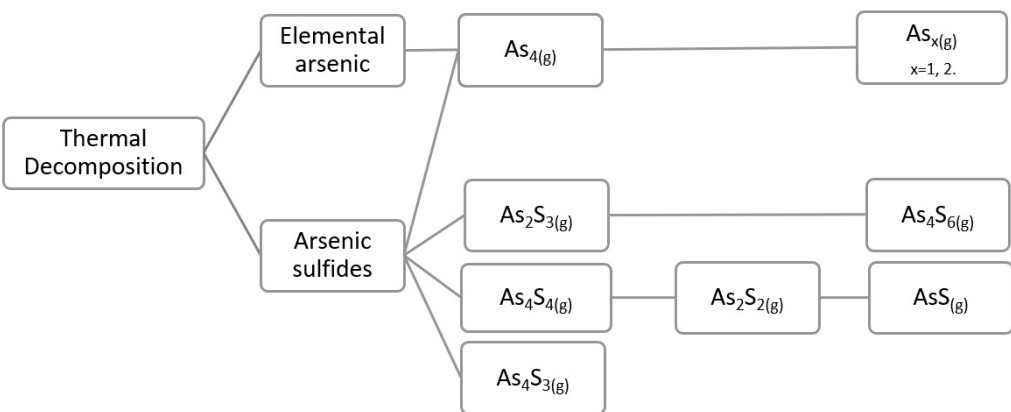

**Figure 1.** Diagram of the predominant volatilizations during thermal decomposition of elemental arsenic and its sulfides in a neutral atmosphere, based on the literature studied. [Own elaboration].

### 2.1. Elemental Arsenic

Studies related to reaction mechanisms during the thermal decomposition of elemental arsenic are very scarce. There is information related to volatilization from copper ores or concentrates during roasting for its removal, as well as studies on arsenic thermodynamic data. Regarding the latter, Rosenbalt (1968) [39] studied the volatilization of arsenic from a high purity arsenic crystal. This crystal was pulverized and treated under an argon atmosphere between 235 °C and 307 °C, using equipment similar to current differential thermal analysis equipment. They obtained a vaporization coefficient that increased with temperature according to the following expression:

$$a_v = \exp(-10{,}921/RT)$$

They indicated that at any given temperature, the vaporization rate remained constant in time, with a slight delay for a short period when the sample begins to heat up, and later proposed that this delay was not due to a temperature adaptation process, but rather that it was linked to an energy barrier; the magnitude of the activation energy found was 184 kJ/mol and the observed vaporization rates implied that vaporization occurred predominantly as $As_{4(g)}$. It was also indicated that there is not enough energy available during vaporization to form $As_{2(g)}$, and the volatilization rate of $As_{4(g)}$ was too high for it to yield $As_{2(g)}$; even if the $As_{2(g)}$ vaporization coefficient was unitary, volatilization as $As_{2(g)}$ would account for less than 5% of the observed weight loss. Therefore, the initial vaporization stage would correspond to the formation of $As_{4(g)}$ in the arsenic structural network, which cannot exist in the $\alpha$-$As_{(s)}$ structure, since $As_{4(g)}$ possesses a tetrahedral arrangement of molecules, whereas the arsenic crystal is rhombohedral; in general, vaporization from solids occurs by a "step-by-step" mechanism in which the atoms or molecules are initially at a transition site in the crystal network, then diffuse along the structure and subsequently become attached to the surface, and are finally desorbed to the gas phase. Therefore, the low vaporization coefficients of arsenic were associated with the rearrangement of bonds during the vaporization process and the existence of gaseous molecular species that could not exist as such in the crystal network.

Rau (1975) [40] studied the gas phase composition for arsenic in order to obtain formation enthalpies and Van der Waals constants for $As_{4(g)}$ and $As_{2(g)}$ species. For this purpose, high purity solid arsenic was used, which was treated in an autoclave by varying the pressure and temperature. It was possible to measure the pressure exerted by a known amount of arsenic enclosed in the predetermined volume of the equipment by using a manometer, which allowed calculating the gas density and saturation vapor pressure. The melting temperature of arsenic was 817 °C and the formation enthalpy of $As_{4(g)}$ and $As_{2(g)}$ were 190 kJ/mol and 152.23 kJ/mol at 25 °C and 1 atm, respectively. Furthermore, it was indicated that the saturated arsenic gas phase at intermediate temperatures consisted

mainly of $As_{4(g)}$. At very high temperatures and/or low $As_{4(g)}$ saturation levels, $As_{4(g)}$ decomposes in two stages to $As_{2(g)}$ and subsequently to $As_{(g)}$.

Lynch (1980) [34] performed a thermodynamic analysis to obtain the Henrian activity coefficient using $As_{(g)}$, $As_{2(g)}$, $As_{3(g)}$, and $As_{4(g)}$ sublimation data from the compiled Hultgren database [35], in order to recalculate an accurate value for the arsenic activity in molten copper. Lynch indicated that there was inconsistency on the arsenic activity coefficient values at different temperatures found in the literature, as they varied in magnitude from $1.45 \times 10^{-4}$ to $5.00 \times 10^{-7}$ at temperatures of 1000 °C and 1300 °C, respectively; to resolve these inconsistencies, he obtained experimental information from several authors on the activity of gaseous arsenic species such as $As_{(g)}$, $As_{2(g)}$, $As_{3(g)}$, and $As_{4(g)}$ volatilizing from molten copper. He found that at 1000 °C, the arsenic activity coefficient was $2.2 \times 10^{-3}$ and at 1373 K it was $5.6 \times 10^{-3}$. However, he did not obtain results at 1300 °C, since they did not have enough values to extrapolate this coefficient.

In parallel, Lau et al. in 1983 [41] indicated that the tabulated values for the reaction rates from the compilation of Hultgren [37] for the equilibrium given by the decomposition of $As_{4(g)}$ to $As_{2(g)}$ used in the Lynch development [34] were erroneous. Lau et al. calculated the arsenic activity by total gas pressures in equilibrium with the molten copper phase. They also developed the experiment for a copper matte dopped with metallic arsenic between 1.04% and 1.6% wt.%. The samples were prepared by melting the elements at 1180 °C in a closed graphite crucible under an argon atmosphere and kept in the molten state for two hours. As a result, the dominant species found was $As_{2(g)}$ and to a lesser extent $AsS_{(g)}$ and $S_{2(g)}$. Subsequently, for an arsenic-free matte sample, which was studied under mass spectrometry between 1027 °C and 1107 °C, only $S_{2(g)}$ was found. The authors concluded that the activities of As and S should be the same in the solid and liquid matte at melting temperature, and that no drastic changes were expected as the temperature of the liquid matte increased. Furthermore, the vapor mass spectra showed that the sulfur potential of the matte was not affected by arsenic, and the observed increase in vapor pressure was associated with the vaporization of $As_{2(g)}$. Therefore, to calculate the arsenic activity, the sulfur pressure was considered to be equal to the pressure of the sample without arsenic. A coefficient for arsenic activity of 0.79 mole fraction$^{-1}$ on average was obtained for an approximate range of 1.04% to 1.6% wt.% between 1012 °C and 1061 °C. Since the activity values obtained were very low, Lau et al. [41] proposed that the most common errors when experimentally determining the arsenic activity coefficient mainly resulted from copper inclusions in matte for the ratio of arsenic distribution in copper analysis. In addition, the iron (Fe) percentage in the sample and the difficulty of correctly saturating the matte with arsenic, since part of the arsenic was lost as it was contained in the walls during cooling by condensation generated errors when measuring the partial pressures when there were minimal amounts of arsenic in the system, causing a large variation in the results obtained for the arsenic activity coefficient. To mitigate the effect of condensed arsenic phases, and to obtain better accuracy, he proposed that it was ideal for future research to work with more than 1% As wt.%.

Subsequently, due to the error found by Lau et al. [43] on the reaction heat of $As_{4(g)}$ dissociation to $As_{2(g)}$, Dabbs and Lynch [34] recalculated the As activity in molten copper based on the values previously proposed by Lau et al. [34] with these new values performed a thermodynamic study and obtained that the predominant gaseous species was $As_{4(g)}$ at 1100 °C. Furthermore, due to this modification, it was obtained that the dissociation rate of $As_{4(g)}$ to $As_{2(g)}$ was faster than previously calculated. It was considered that the calculations were corrected assuming that $As_{3(g)}$ was not present since this allotropy had not been observed experimentally by spectrometry and presented a very high pressure. The effect of the presence of $As_{3(g)}$ on the results was calculated, and there was only a slight increase in the partial pressures of $As_{2(g)}$ and $As_{4(g)}$ between 650 K and 800 K. Finally, they obtained that the activity coefficient at 1000 °C was $3.98 \times 10^{-3}$ and $2.63 \times 10^{-3}$ at 1100 °C.

Gokcen in 1989 [33], based on experimental data from various sources, studied the thermodynamic behavior of the As system, focusing on the solid–liquid, solid-gas, and gas

phase equilibria. By ordering the existing experimental data, he obtained enthalpies and entropies for the mentioned phases and equilibria. He indicated that the only elemental solid phase of arsenic stable above 70 kbar was αAs, and that the gas phase was composed of $As_{(g)}$, $As_{2(g)}$, $As_{3(g)}$, and $As_{4(g)}$, with $As_{4(g)}$ as the predominant phase (approximately 94%) for a temperature of 1127 °C; furthermore, at 614 °C the gas phase was composed mainly of $As_{4(g)}$, together with $As_{(g)}$, $As_{2(g)}$, and $As_{3(g)}$ in concentrations close to 0.19% v/v. These percentages were lower at lower temperatures.

To summarize the information provided, it can be inferred that volatilizations during thermal decomposition of elemental arsenic are mainly given by the $As_{4(g)}$ phase, however, as the temperature increases, it decomposes in the form of its allotropic $As_{(g)}$, $As_{2(g)}$, and $As_{3(g)}$; however, only $As_{(g)}$ and $As_{2(g)}$ are the predominant ones as can be seen in Figure 1.

### 2.2. Arsenic Sulfides

According to the literature reviewed, in a neutral atmosphere, arsenic sulfides usually decompose to the most stable phase according to the operating conditions, whether atmosphere, gas concentration, heating rate, and temperature, among others.

Szarvasy and Messinger [42] (1897) studied the arsenic compounds $As_2S_2$ and $As_2S_5$ at high temperatures, in the range of 388 °C to 1200 °C. The analysis focused on the behavior of these sulfides at high temperatures using vapor density determination. The authors emphasized that at that time (1897) only the following arsenic sulfide species, $As_2S_5$, $As_2S_3$, $As_2S_2$, and $As_4S_3$, were known, and that the system was very complex to study. According to the authors, when evaluating a sample of $As_2S_5$, it decomposed at 500 °C into $As_2S_{3(g)}$ and $S_{2(g)}$, whereas this sulfide formed remained stable at 700 °C, and dissociated at 1000 °C; however, the authors did not report the dissociation products; they also emphasize that, at temperatures below 550 °C, $As_4S_{4(g)}$ remained stable. At temperatures above 550 °C, $As_4S_{4(g)}$ decomposed to form $As_2S_{2(g)}$. $As_2S_{2(g)}$ was studied extensively at different temperatures, and by measuring the vapor density at 900 °C they obtained a value equal to $\rho = 7.403$, which remained constant up to 1100 °C, this density value matches with the tabulated values for $As_2S_2$. The authors add that at temperatures above 1200 °C another dissociation occurs, without indicating the type. Finally, they conclude that the stability of arsenic sulfides in the temperature range studied depends directly on the sulfur stoichiometry present in the molecule, since the lower the number of sulfur atoms associated with arsenic, the greater the thermal stability of the sulfides.

Szarvasy et al. in 1897 [44], propose the possibility of an $As_4S_{5(g)}$ species, which appears not to have been reported previously. They also emphasize that although $As_4S_4$ is reported in the literature to transform to $As_2S_3$, they did not find the presence of $As_4S_6{}^+$-dimer formed in the $As_2S_3$ volatilization phase, which was reported later by Lu and Donohue in 1944 [43]. The sublimation pressure for $As_4S_4$ was determined for equilibrium and pressure conditions under the free-surface form. For the equilibrium case, sublimation was determined between 179 °C and 267 °C, yielding the expression:

$$Log(P) = (8.021 \pm 0.200) - ((0.619 - 0.010 \times 10^4)/T)$$

The heat and entropy of sublimation at 220 °C were $117.15 \pm 2.1$ kJ/mol and $36.7 \pm 0.9$ eu, respectively. The results obtained for the sublimation of the realgar mineral free-surface pressure for the temperature range between 138 °C and 192 °C, were:

$$Log(P) = (8.091 \pm 0.620) - ((0.637 - 0.027 \times 10^4)/T)$$

The reaction heat was $121.75 \pm 5.0$ kJ/mol, and the sublimation entropy was $37.0 \pm 2.8$ eu.

Subsequently, in 1944, Lu and Donohue [43] developed an electron diffraction-based study to study the structure of $As_4S_4$, $As_2S_3$, $N_4S_4$, and $S_2$. The arsenic mineral samples were purchased from a supplier in Pasadena, the realgar mineral came from the White Caps mine in Manhattan, Nevada, and the orpiment from Mercur, Utah. The realgar

mineral sample was purified in a vacuum at a temperature of 300 °C, and the orpiment was treated at the same atmospheric conditions between 300 °C and 400 °C to remove possible impurities and form an $As_2S_3$ crystal which was subsequently pulverized. In addition, they artificially prepared a high-purity $As_2S_3$ sample by precipitation by mixing $H_2S$ and $AsCl_3$ in HCl. To determine the structure of the arsenic sulfides, they used high-temperature electron diffraction microscopy, obtaining the $As_4S_{6(g)}$ dimer as the predominant volatile element from $As_2S_3$; this dimer has the same molecular structure as $As_4O_{6(g)}$-tetrahedral, due to the fact that the molecules have similar bonds and angles of magnitude. For the case of $As_4S_4$ and $N_4S_4$, they were not able to determine with certainty the crystal structure, however, they concluded that the vertices of the $As_4S_4$ structure are occupied by arsenic atoms instead of sulfur atoms, and these arsenic atoms are connected by single bonds.

In 1971, Munir et al. [37] studied the sublimation of $As_4S_4$ using mass spectrometry and vapor pressure measurement, further stated that the $As_4S_4$ molecules were held together by Van der Waals forces; emphasizing the study of Lu and Donohue [43], they mention that the shapes and dimensions of the gas phase molecules are similar to the solid phase. Munir et al. [37] In their experimental work they used a natural and a synthetic sample of $\beta$-$As_4S_4$ (orthorhombic), which was the result of exposing $\alpha$-$As_4S_4$ (monoclinic) to sunlight; the samples were melted at 127 °C in a vacuum and analyzed by spectrometry, obtaining, as a result, the presence of $As_4S_4^+$, $As_4S_3^+$, $As_3S_3^+$, $As_4^+$, $As_3S^+$, $As_3S^+$ or $S_8^+$, and $As_2^+$ for both samples. Due to the low potentials obtained in the spectrometry of $As_4S_{4(g)}$, i.e., low amount of $As_4S_4^+$, it was suggested that $As_4S_{4(g)}$ dissociated in multiple forms, and the most thermodynamically favorable dissociation would be in the form of $As_4^+$ and $S_4^+$; however, only the $As_3S_3^+$ species could be attributed to this phenomenon since the measured amounts were minimal, whereas $As_2^+$, $As_4^+$, and $As_4S_3^+$ are found in higher amounts, and would probably correspond to the $As_4S_{4(s)}$ partial thermal decomposition; therefore, they concluded that the $As_4S_{4(s)}$ sublimation was dissociative in nature. Furthermore, the authors concluded that the predominant phases for temperatures below 127 °C were $As_4S_3^+$, $As_4^+$, and $As_2^+$, and as the temperature was increased, the magnitude of the $As_4S_4^+$ phase increased, reaching 50% of the gas phase at 127 °C, and at 227 °C the amount of $As_4S_{4(g)}$ corresponded to 75% of the gas phase; it could reach 90% if a possible error in the detection of the volatilized species was assumed.

Rogstad in 1972 [44], studied by Raman spectrometry a mixture of elemental arsenic and sulfur vapors at 800 °C. The author, prior to his experimental work, stated that several species have been found for the As-S system, for example, for the case of $As_4S_4$ this can be in the $\alpha$ (monoclinic) or $\beta$ form, whereas $As_4S_3$ in $\alpha$ form, and unlike $As_2S_3$ is polymeric and is considered to be layered. However, $As_4S_4$ upon exposure to light is readily converted to $As_2S_3$. The authors refer to the study of Szarvasy (1897) [42], stating that $As_4S_4$ is stable below 550 °C and at higher temperatures dissociates into $As_2S_2$; they also mention the study of Munir et al. (1971) [37], where the formation of $As_3S_3^+$ is due to the ionization of $As_4S_{4(g)}$, and that the dominant species at low temperature were $As_4S_3^+$, $As_4^+$, and $As_2^+$, and temperatures above 227 °C the amount of $As_4S_{4(g)}$ corresponded between 75% and 90% of the gas phase, which decreases with increasing temperature. For their experimental tests, they used a mixture of arsenic and pulverized sulfur, with different $As_4$:$S_8$ ratios, conditions of excess arsenic (10:1–3:1), excess sulfur, and approximately equimolar amounts (3:2–1:1) in the range 480 °C to 900 °C. For the sulfur-rich sample, several gaseous sulfur allotropies were found, and for the arsenic-rich sample, a very complex spectrum was obtained, indicating the presence of multiple species. For the As:S ratio of 10:1, it was observed in the temperature range between 480 °C and 610 °C that the gas phase is mainly composed of $As_4S_{3(g)}$ and $As_{4(g)}$. For the 3:1 ratio, the spectra obtained by spectrometry did not change significantly with temperature between 550 °C and 770 °C, and the presence of $As_4S_{4(g)}$ together with $As_4S_{3(g)}$ and $As_{4(g)}$ was observed. The sample with 3:2 ratio had $As_4S_3$ as the dominant phase between 570 °C and 650 °C. Subsequently, at temperatures above 700 °C $As_2S_2$ was obtained. Finally, they concluded that they cannot provide a detailed analysis of the species present in the system, however, they can state that the volatilizations contain

$As_4S_{4(g)}$, or $As_4S_{4(g)}$ mixed with $As_4S_{3(g)}$ depending on the experimental conditions; and that the single existence of the $As_4S_{4(g)}$ species from $As_4S_{4(s)}$, and $As_4S_{6(g)}$ from $As_2S_{3(s)}$, was doubtful since there are many species involved.

In 1978, Janai et al. [45] studied volatilizations from a sample of $As_2S_3$ prepared by a vacuum sublimation of an orpiment mineral pulverized sample from BDH Chemical Ltd., which was vacuum sealed in a Pyrex® tube and then heated at 550 °C for 10 h. This yielded $As_2S_3$ crystals, part of which were volatilized in a vacuum and subsequently cooled on an aluminum plate, thus obtaining two samples: one of $As_2S_3$ crystals and one $As_2S_3$ film, both of which were pulverized and analyzed by mass spectrometry, using the Atlas MAT CH4 equipment in a vacuum between 60 °C and 400 °C. The author, based on data reported by K.C. Mills (1974), in the database "Thermodynamic Data for Inorganic Sulfides Selenides and Tellurides" [35], mentions that the known primary volatilizations of $As_2S_{3(s)}$ are AsS and $S_2$, or $As_nS_n$ and $S_2$, with 1 < n < 4; and that $As_4S_{6(g)}$ has been reported as a minor constituent of the gaseous phase, whereas the predominant volatilized phases are AsS, $As_4S_4$, $As_4$, and $As_2$. The authors only reported their mass spectrometry results for the volatilizations obtained at 300 °C, for the glass sample, since the results for the film were similar. A large number of arsenic gas phases were obtained; they found all possible species for $As_mS_n^+$, where 0 < m < 4 and 0 < n < 5, with the exception of $S_4^+$, $S_5^+$, $AsS_5^+$, $As_3S_5^+$, $As_3S_5^+$, $As_4S_2^+$, and $As_4S^+$. In addition, $As_4S_6^+$ was not observed, and if present it should be found in a proportion below the detection limits of the technique used. The authors concluded that at temperatures below 400 °C the species that volatilized was $As_4S_{6(g)}$; and that, the absence of $As_4S_6^+$ and presence of gaseous ionic compounds, suggested that arsenic volatilized mainly as $As_4S_{6(g)}$, then became unstable and ionized, to later fractionate into gaseous ions; such as $As_4S_5^+$ and minor ions, which is an unknown molecular form of arsenic sulfides.

Later in 1980, Johnson et al. [31] studied the formation of enthalpies and thermodynamics of the species $As_4S_4$ and $As_2S_3$, since the authors found a discrepancy in the thermodynamic data of these compounds reported up to this date (1980), since in the formation enthalpies of $As_4S_4$ and $As_2S_3$ there are differences of up to 140 kJ/mol and 75 kJ/mol, respectively. In addition, due to the difficulty of obtaining pure and natural $As_4S_4$ and $As_2S_2$ for the development of calorimetry tests, it was necessary to prepare these compounds in the laboratory, which did not have the same crystalline form as the natural one. $As_4S_4$ at 280 °C presents an irreversible transformation from α-monoclinic- to β-orthorhombic-phase, the latter being the artificially produced phase. For the case of $As_2S_3$, a non-crystalline glassy material was obtained. The authors assume that it is possible to predict the properties of the natural form with synthetic materials of high purity. The studied samples of $As_4S_4$ and $As_2S_3$ were prepared by reacting high purity As and S in a quartz vacuum tube which was kept at 400 °C for several days, then the samples were melted in a vacuum to remove possible impurities, such as $As_2O_3$, which was highly volatile. Six experimental calorimetry tests were developed for each sulfide in the temperature range between 101 and 576 °C under a helium atmosphere. They obtained that the standard formation enthalpy at 25 °C is $-134.6 \pm 6.7$ kJ/mol and $-69.6 \pm 4.2$ kJ/mol for β-$As_4S_{4(s)}$ and $As_2S_{3(glassy)}$, respectively. In addition, they developed a theoretical study with data obtained from several authors, including Munir et al. (1971), and determined the standard formation enthalpy at 25 °C obtaining values of $-138.1 \pm 6.7$ kJ/mol and $-91.6 \pm 4.2$ kJ/mol for α-$As_4S_{4(s)}$ and $As_2S_{3(s)}$, respectively.

In 1986, Hamman and Santiago [46] emphasized the great interest in the study of $As_4S_6$ properties, analyzing by mass spectrometry the volatilizations from two samples of $As_4S_6$, prepared in different ways in order to compare the volatilizations according to the way of crystal production. The first sample was prepared in a vacuum inside quartz tubes at 827 °C for 15 h, allowing the tube to cool down to room temperature to obtain a glassy sample, whereas the second was prepared by evaporation of a solution in Servofrax glass by spin coating. The samples were studied under vacuum conditions $-10^{-7}$ torr- at 190 °C and a sample mass varying between 28 and 800 mg. They obtained all the gaseous

arsenic sulfide phases known to date—AsS, $As_2S_2$, $As_3S_3$, $As_4S_4$, and $As_4S_6$—and a variety of previously unknown phases, such as $As_4S_2$ and $As_4S_5$. For the sample obtained by the first technique $As_4S_{6(g)}$ was obtained, whereas for the second technique there was no presence of this molecule, the authors suggested that in these samples there was a high density of $As_4S_{6(g)}$ that during spectrometry is ionized, becoming unstable, subsequently fractionated and dissociated into several species, and therefore, due to this situation it is often not detected as a product. They also conclude that due to the different techniques used to obtain the samples, and different cooling times, the magnitude of the volatilizations could depend directly on the thermal history of the samples studied. Finally, they propose that $As_4S_{6(g)}$ is found in its stable form as a minor constituent before the system reaches equilibrium and that AsS, $As_2S_2$, $As_3S_3$, $As_4S_4$, $As_4S_5$, and $S_2$ are the predominant species in the gas phase.

In 1999, Nakazawa et al. [30] studied the removal of arsenic sulfides from copper concentrates, since the arsenic behavior and optimum conditions for its removal are not evident, due to the wide variety of arsenic species present and the phases in which they are found, due, in turn, to the arsenic chemical activity, as well as to its different valences and affinity for different elements (sulfur and oxygen, among others). Given the nature of arsenic, its species are characterized by being highly volatile, since they have a high vapor pressure, and remain in the concentrated solution as a non-stoichiometric product. The authors first developed their study with equilibria using information compiled in HSC chemistry software's databases, now owned by Outotec [47], and consider that calcine is a pseudo solid solution, where the arsenic-containing species are dissolved in that phase. To perform these calculations in HSC chemistry [47] they used three fictitious samples containing 1000 moles of $CuFeS_2$ and 100 moles of $FeS_2$, varying the arsenic content by adding 100, 10, and 1 mole FeAsS, and 27, 2.7, and 0.27 moles of $As_2S_3$, respectively. Depending on the mass percentage of arsenic used, arsenic was added to the FeS system in order to keep the amount of sulfur in the system constant. The authors mention that in reality arsenic is contained in minor amounts in the form of $Cu_3AsS_4$ and $Cu_{12}As_4Sb_{13}$, however, due to the lack of thermodynamic information on these species, these compounds were omitted. The authors varied the oxidation degree by performing their simulation from 1000 to 4500 moles of $O_2$, for a low degree of oxidation—which can be approximated to a neutral atmosphere—the predominant volatilized species were elemental and sulfur phases, such as $As_2S_{3(g)}$, $As_4S_{4(g)}$, $As_{4(g)}$, and $As_{2(g)}$, whereas for high degrees of oxidation these were gradually replaced by $As_4O_{6(g)}$. In addition, with the results obtained, they simulated how the content of a fictitious sample with 5.3% total arsenic varied during roasting at 700 °C, finding that its removal depends directly on the temperature and the degree of oxidation used and that the lower the amount of arsenic present in the system, the more difficult its removal by roasting will be.

From the data analyzed for the simple arsenic sulfides' decomposition, we can conclude that arsenic sulfides volatilize in multiple phases in parallel; but the predominant gaseous phases are $As_4S_6$, $As_4S_4$, $As_4S_3$, and $As_4$, shown in Figure 1. Additionally, according to Hammn et al. [46] the magnitude of these volatilizations would depend directly on the thermal history of the samples studied, i.e., the previous conditions to which they were subjected before the study.

Additionally, we can confirm Szarvasy et al. [44] who conclude that the lower the number of sulfur atoms associated with arsenic, the higher the thermal stability of the sulfides, which is clearly seen in the stability of $As_4S_{6(g)}$ and its dissociation to multiple sulfides of lower stoichiometry; and in the species $As_4S_{4(g)}$, which at high temperatures dissociates in two stages to $As_2S_{2(g)}$ and $AsS_{(g)}$. Furthermore, although the $As_4S_{6(g)}$ dimer is often not detected, it is present as the predominant phase. Both Janai et al. [45] and Hamman et al. [46] agree that the $As_4S_{6(g)}$ phase is the predominant phase; however, it is not detected because it ionizes and dissociates into other species when analyzed spectrometrically.

### 2.3. Arsenic Sulfide Compounds

Due to the scarce information regarding the thermal decomposition of arsenic sulfides, it was necessary to study the mechanisms of arsenic volatilization from compound or mixed sulfides such as enargite, where arsenic is mainly in its pentavalent state. This analysis can be of great help to understand and develop the mechanisms and reaction kinetics of arsenic sulfides because enargite is one of the main minerals with high arsenic content in Chilean metallurgy; in turn, similar to most sulfides, it is inherently unstable in exogenous media and decomposes by roasting at high temperatures.

In 2001, Padilla et al. [26] studied the thermal decomposition of enargite under a nitrogen atmosphere in order to determine the reaction mechanisms and the general kinetics of enargite thermal decomposition. The enargite sample studied was obtained from El Indio gold mine of Barrick Corporation, Chile. The sample had a composition of 17.6% As, 45.8% Cu, 31.7% S, and 47 ppm Fe. Based on this composition, and assuming that arsenic was only present in the enargite, the sample was considered to contain 92.4% enargite, which was corroborated by X-ray diffraction (XRD), obtaining that the major component was enargite, and the major impurity was quartz. The decomposition rate of $Cu_3AsS_4$ was determined, by means of weight loss as a function of time; the tests were carried out in thermogravimetric equipment containing a quartz tube hanging vertically from a balance towards the center of the furnace and a gas control system. The sample quantity used was 100 mg and had an average particle size distribution between $-45$ to $+38$ µm; the temperature range studied was 575–700 °C. The decomposed enargite fraction was determined by weight loss with respect to the initial sample, and the remaining product was characterized by XRD. From a sample of grain size distribution from $-75$ to $+53$ µm, studied under non-isothermal conditions, with a heating rate of 5 °C/min, it was obtained that the enargite started to decompose at 550 °C. In addition, they obtained that the enargite decomposed slowly at 550 °C, and subsequently increased its decomposition rate above 575 °C. The results obtained for the mole fraction between $-45$ to $+38$ µm. Where it can be seen how the temperature affects the decomposition rate, it is also possible to visualize that below 650 °C, between 4 and 8 min there is a decrease in the decomposition rate, which would indicate that the enargite decomposition occurs through the intermediate formation of compounds; above 650 °C the change in the curve is not appreciable.

The authors reported the possibility of two reaction mechanisms, one with the intermediate formation of covellite and the other with tennantite, regardless of which of the two pathways the decomposition proceeds in the arsenic removal stage and is released as gaseous $As_4S_4$; however, it was not possible to identify exactly whether the volatilization was in the form of $As_4S_4$ or $As_2S_3$. $As_{4}S_{4(g)}$ was the form chosen because it was thermodynamically more stable and feasible for the conditions studied, according to the data provided by the following reactions at 700 °C proposed by the author.

$$2Cu_3AsS_4 = 3Cu_2S + As_2S_{3(g)} + S_{2(g)} \ K^{700°C} = 6.10 \times 10^2$$

$$4Cu_3AsS_4 = 6Cu_2S + As_4S_{4(g)} + 3S_{2(g)} \ K^{700°C} = 1.55 \times 10^4$$

$$4Cu_3AsS_4 = 6Cu_2S + As_{4(g)} + 5S_{2(g)} \ K^{700°C} = 1.17 \times 10^{-1}$$

According to the values of the equilibrium constants ($K^{700°C}$), the value of $1.55 \times 10^4$ is the highest, which would indicate that arsenic would volatilize as $As_{4}S_{4(g)}$ from enargite at that temperature.

Subsequently, to verify the identity and formation of the intermediate compounds during the thermal decomposition of enargite, partially reacted samples at 575 and 650 °C were analyzed by XRD. The species obtained by XRD at 575 °C were unreacted enargite, tennantite, and chalcocite, whereas at 650 °C, tennantite was identified as the predominant phase, together with minor digenite and chalcocite phases. These results showed that the enargite at these temperatures is decomposed by the formation of tennantite, chalcocite, and non-stoichiometric copper sulfides ($Cu_{1.8}S$ and $Cu_{1.96}S$). On the other hand, at 700 °C

the enargite starts to decompose in 2.5 min, and only tennantite, chalcocite, and non-stoichiometric copper sulfides ($Cu_{1.8}S$ and $Cu_{1.96}S$) were identified; at 5 min the same phases were identified, but tennantite decreased in magnitude and copper sulfides increased; at 15 min, only chalcocite and non-stoichiometric copper sulfides were found. These results suggest that the decomposition of enargite above 700 °C is very rapid.

Considering the results obtained and the thermodynamic stability of $As_4S_{4(g)}$, the following two-step reaction mechanism is proposed.

$$(Cu_3As_4)_4 = Cu_{12}As_4S_{13} + 1.5S_{2(g)} \tag{1}$$

$$Cu_{12}As_4S_{13} = 6Cu_2S_{1-x} + As_4S_{4(g)} + (1.5 - 3x)S_{2(g)} \tag{2}$$

where x, varied between 0 and 0.11, depending on the copper sulfide species. Thus, obtaining the following general reaction for the enargite decomposition in neutral atmosphere.

$$(Cu_3As_4)_4 = 6Cu_2S_{1-x} + As_4S_{4(g)} + (1.5 - 3x)S_{2(g)}$$

The calculated activation energies were 125 kJ/mol for the first reaction and 236 kJ/mol for the second reaction. These values indicate that the enargite decomposition is controlled by surface reaction.

In 2004, Winkel et al. [48] studied the enargite decomposition and a copper concentrate by thermogravimetry and inductively coupled optical emission spectroscopy (TGA-ICP). For this purpose, enargite was synthesized from copper wire, $As_2S_3$ crystals, and sulfur. The copper concentrate used was dried at 60 °C for 14 h, and then taken to a ball mill for 10 min to homogenize the sample. The major mineral species in the concentrate were chalcopyrite, pyrite, pyrrhotite, and quartz.

For TGA-IPC analysis, 8 to 12 mg heated above 960 °C at a heating rate of 4 °C/min, in a Mettler Toledo TGA/SDTA 851 apparatus, under 80 mL/min of argon (99.999%) were used. The gases released were simultaneously analyzed by inductively coupled plasma optical emission spectrometer (ICP-OES). The solid product was analyzed by XRD on Philips X-Pert equipment.

The results obtained for enargite shows the weight losses and volatilizations of arsenic and sulfur detected during the thermal decomposition of enargite. The ICP results show 5 peaks for sulfur and 2 for arsenic. Between 581 and 638 °C, a large weight loss occurs, corresponding to sulfur and arsenic emissions, these peaks are consistent with volatilization in the form of $As_4S_{4(g)}$, whereas in the solid phase there is the presence of chalcocite accompanied by small amounts of digenite, and traces of elemental copper. Considering that all the arsenic is volatilized and that only chalcocite remains in the solid sample, a weight loss of 39.4% is theoretically expected, compared with the experimental one which was 38.4%; this 1% variation may be due to the formation of digenite. In addition, the multiple sequential peaks during sulfide and arsenic evaporation suggest the formation of intermediate compounds.

The results obtained for the copper concentrate which is divided into two graphs, one for TG-DTG and the other for the results of gas analysis by ICP. Between 476 and 568 °C, the highest weight loss occurs due to the volatilization of 55% of the total sulfur; in addition, two arsenic volatilizations occur at temperatures of 517 and 568 °C. Peak volatilizations of Zn, Pb, and Cd were detected above 900 °C, and it was emphasized that sulfur volatilizations did not increase at this temperature.

In 2008, Winkel et al. [49] studied the thermal decomposition of three copper sulfides: chalcocite, chalcopyrite, and enargite; and two commercial copper concentrates in the range of 900 to 1500 °C in an inert atmosphere. The two copper concentrates were obtained from Outokumpu Oy, Finland; they contained 0.03 and 0.86% copper and were characterized by 33.4 and 36.4 wt.% sulfur, respectively. Chalcocite was obtained from Sigma Aldrich. Chalcopyrite was synthesized from the elemental phases in a vacuum at 1200 °C for 30 min. Enargite was synthesized in a vacuum from copper wire, $As_2S_3$ crystals, and mineral sulfur at 752 °C, and subsequently quenched at 500 °C for 14 days. Tests were carried out in the

range of 900 and 1500 °C in a Netzssch STA 409 thermobalance under an inert atmosphere of $N_2$ (99.995%) or a CO-CO$_2$ mixture. A Mettler Toledo thermobalance coupled to ICP-OES equipment was used to analyze the gases. As for the concentrates studied, both behaved similarly, varying in the magnitude of the mass lost, which was a direct consequence of the difference in the sulfur content of each sample. In the case of chalcopyrite and chalcocite, they obtained the following reaction mechanisms based on their experimental results, respectively.

$$CuFeS_2 = 1/5Cu_5FeS_4 + 4/5FeS + 1/5S_2$$

$$Cu_2S = Cu_{(l)} + 0.5S_{2(g)}$$

The results obtained by TG-ICP for the enargite can be seen in Figure 2, where a high concentration of sulfur and arsenic in the gas phase can be appreciated, which were released mostly at temperatures between 527 and 677 °C. Sulfur started to volatilize at 500 °C and four peaks ($S_1$, $S_2$, $S_3$, and $S_4$) were obtained, the highest peaks $S_1$ and $S_2$ occurred at 854 and 911 K, respectively. Two arsenic peaks are registered: the lowest (A1) at 581 °C, and the highest (A2) at 630 °C. The S1 peak corresponds to the formation of tennantite according to the following reaction, which has a theoretical weight loss of 15.5%, although the actual measured value was 25.5%.

$$4Cu_3AsS_4 = Cu_{12}As_4S_{13} + 1.5S_{2(g)}$$

The $S_2$ peak coincides with the second arsenic peak ($A_2$) at 638 °C, which is in agreement with the formation of chalcocite proposed by Padilla et al. [26], represented in the following reaction.

$$Cu_{12}As_4S_{13} = 6Cu_2S_{1+x} + As_4S_{4(g)} + (1.5 - 3x)S_{2(g)}$$

Although the $S_1$ peak coincides with $A_1$, which would represent a slight volatilization of $As_4S_{4(g)}$ at 547 °C.

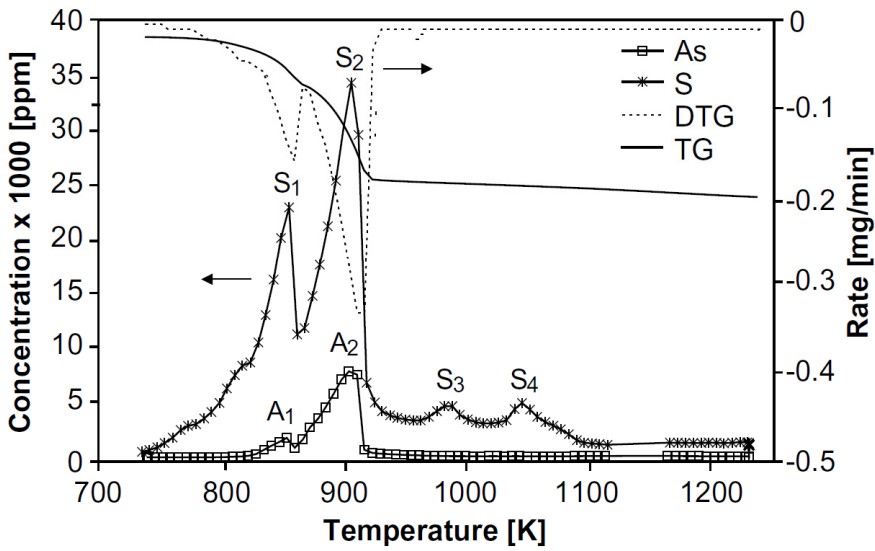

**Figure 2.** TG-ICP analysis for enargite [49].

ICP analysis for arsenic volatilized from enargite resulted in $As_4S_{4(g)}$ decomposing at 827 °C into $S_{2(g)}$ and $As_{4(g)}$, where the $As_4$ formed becomes unstable at 1200K and splits into $As_{2(g)}$. The solid phase analyzed by XRD results in the formation of chalcocite and a minor digenite phase with traces of elemental copper; in addition, this analysis shows the absence of arsenic above 727 °C; therefore, total volatilization of this element is obtained.

In 2013, Wilkomirsky et al. [50] studied the physico-chemistry of an arsenic-bearing copper concentrate by partial roasting. The samples studied came from División Ministro Hales (DMH) of CODELCO, the sample composition was 30.1% $Cu_2S$, 24.4% $FeS_2$, 21.0% $Cu_3AsS_4$, and 7.3% $CuFeS_2$, mainly.

The concentrate was roasted in a fluidized bed furnace at 680–725 °C under an oxidizing atmosphere, with enough air to oxidize the sulfur and generate $SO_{2(g)}$ and an excess of oxygen to generate at most 5 to 6% of magnetite in the calcine, because an excess of magnetite is harmful to the subsequent stages, also a higher oxygen potential can form hematite, which could generate arsenates. It is also desired to oxidize between 1 and 2% of $As_2S_{3(g)}$ to $As_2O_{3(g)}$ to decrease the arsenic content present.

Above 600 °C tests were carried out under a neutral or slightly oxygenated atmosphere, and thermal decomposition of enargite, covellite, and pyrite occurred forming chalcocite and pyrrhotite as solid by-products, as well as $As_2S_{3(g)}$—in the form of $As_4S_6$—and $SO_{2(g)}$ in the gaseous phase. The reactions posed as the reaction mechanism were as follows.

$$2Cu_3AsS_4 = As_2S_{3(g)} + 3Cu_2S + S_{2(g)}$$

$$2CuS = Cu_2S + 0.5S_{2(g)}$$

$$FeS_2 = FeS + 0.5S_{2(g)}$$

$$S_{2(g)} + 2O_2 = 2SO_{2(g)}$$

$$3FeS_2 + 5O_2 = Fe_3O_4 + 3SO_{2(g)}$$

$$As_2S_{3(g)} + 4.5O_{2(g)} = As_2O_3 + 3SO_{2(g)}$$

Finally, they proposed a thermodynamic and kinetic model to explain the formation of bornite, chalcopyrite, and magnetite during partial roasting of copper concentrates containing enargite. They conclude that the proposed reaction mechanism is valid, involving gas-solid reactions between chalcocite, pyrrhotite, and sulfur. At 700 °C the thermal decomposition of enargite, covellite, and pyrrhotite occur very fast. Additionally, it is possible to remove much of the arsenic in the concentrate (1–2%).

We can deduce from the information studied that the volatilizations of arsenic during the thermal decomposition of complex arsenic sulfides, such as enargite, are similar to those obtained from simple arsenic sulfides shown in Figure 1, being present in the phases $As_4S_{4(g)}$ [26,48,49] which decompose at high temperatures into $As_2S_{2(g)}$ and $AsS_{(g)}$; and, $As_2S_{3(g)}$ which forms the $As_4S_{6(g)}$ dimer [50] in the absence of oxygen.

## 3. Oxidation

In an oxidizing atmosphere, the elemental phases and their sulfides oxidize in the presence of oxygen. This oxidation depends mainly on the operating conditions; in the case of oxidation processes, the variable that has the greatest impact is the partial pressure of oxygen and sulfur, which directly influences the stability of the arsenic phases. According to the information analyzed [20–57], the reaction mechanisms during the oxidation of arsenic—elemental and some of its sulfides—in the oxidizing atmosphere are summarized in Figure 3, which will be discussed later.

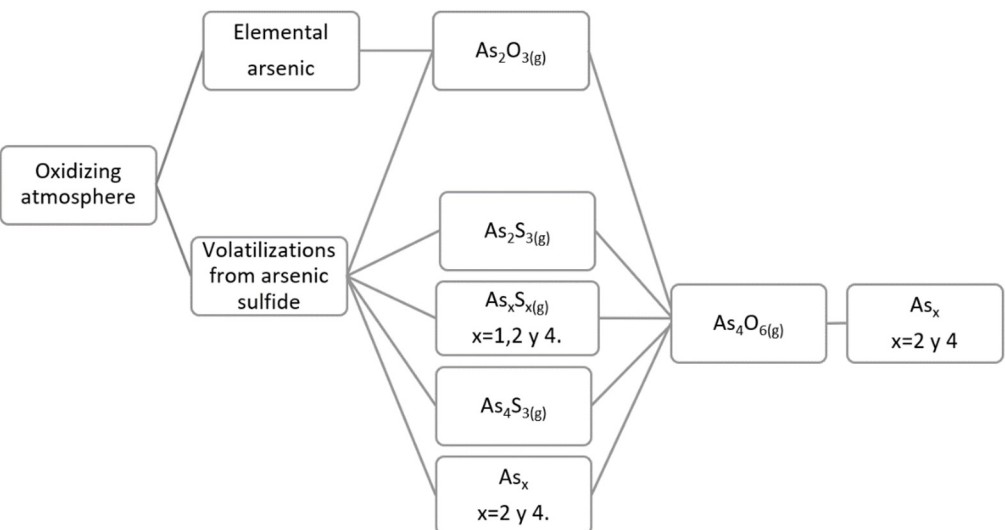

**Figure 3.** Volatilization scheme from the oxidation of elemental arsenic and its sulfides in an oxidizing atmosphere, based on the literature studied. (Own elaboration).

### 3.1. Elemental Arsenic

Studies related to the reaction mechanisms during thermal decomposition and oxidation of elemental arsenic are very scarce. However, the existing information is related to the volatilization and oxidation of arsenic from arsenical minerals or copper concentrates treated under pyrometallurgical processes.

In 1979, Weisenberg and Bakshi [22] analyzed arsenic removal in copper smelting for different types of reaction furnaces, intending to organize the information present to date on the distribution and control of arsenic in the main copper smelting systems in the United States; there was no experimental work. The authors emphasized the scarcity and inconsistency of the information present on arsenic distribution in the literature analyzed, provided by The Environmental Protection Agency; of the information that was studied by the authors they emphasize the amount of arsenic in the feed, since in most U.S. smelters it was generally below 1.0 wt.%, and this value was not constant; therefore, due to the small amount of arsenic present in the feed, in many cases, it cannot be adequately measured. In addition, they noted that changes in smelting techniques and equipment tend to produce variations in arsenic distribution; and, that the presence of other elements such as oxygen or copper within the smelting process can directly influence arsenic distribution. According to the authors, arsenic in copper smelting was removed by volatilization in elemental or sulfide form (specific phases are not reported), and subsequently oxidized in the presence of oxygen to gaseous $As_2O_3$ which upon cooling condenses and is collected as residual dust. However, in an oxidizing atmosphere, the trioxide formed oxidizes to $As_2O_5$, which is less volatile and forms non-volatile arsenates of calcium or iron.

For the distribution and control of arsenic in different reactors, the authors mention that the sulfides present in the copper concentrate that need to be oxidized by fluidized bed roasting are the FeS, $As_2S_3$, $As_2S_3$, and $As_2S_5$ phases; and the arsenic removal was by its volatilization which depends on the temperature, residence time and the type of atmosphere. According to the authors, in order to explain the temperature used during roasting, they mention that $As_2S_3$ has its boiling point at 707 °C, whereas $As_2S_5$ sublimates at 500 °C through decomposition; meanwhile, $As_2O_3$ melts at 310 °C and boils at 457 °C. Consequently, if copper concentrates are heated above 700 °C, arsenic removal should be assured. However, the copper concentrate has a large amount of sulfur, and high temperatures, favorable for arsenic removal, are not consistent for sulfur removal, since at the beginning of roasting a large part of easily meltable sulfides are present, a high temperature would melt the particles, which would reduce the surface-to-volume ratio for sulfur removal; however, if a low temperature—not specified by the author—is

maintained during the whole roasting process, sulfates would be formed, because the sulfates' formation heat would be higher than that of their oxides; for example, for the case of $FeSO_4$ and $Fe_3O_4$ formation from FeS at 400 °C, heat of formation value is $-835.25$ kJ and $-1731.87$ kJ, respectively [47]. Therefore, the ideal condition proposed by the authors was to start with low temperatures and gradually increase up to the roasting temperature. Finally, to conclude their discussion on roasting, the authors propose that arsenic removal by reductive roasting would be feasible because the maximum As removal takes place between 498 and 593 °C under reductive conditions; however, oxidative roasting is required to remove sulfur. Therefore, alternating between oxidizing and reducing processes several times during roasting could maximize the volatilization of arsenic and sulfur, which can be developed in a multi-deck roaster as opposed to a fluidized bed that allows only one of these conditions.

At the date of publication, 1979, most U.S. smelters were using the reverberatory furnace for copper smelting, but the trend in newer or modified smelters is to smelt instantaneously, in molten bath (Noranda, SKS, Ausmelt, Teniente Converter, others) or electric furnace smelting. This change results in reduced production costs and increased $SO_2$ concentration for environmental control; in the reverberatory furnace, arsenic is mainly removed by volatilization and as a slag constituent. If the arsenic fed is greater than 0.2%, it volatilizes between 55 and 75%, and 10–25% of it is slagged, whereas if arsenic is present in smaller quantities, it volatilizes between 5 and 37% and is slagged between 16–55%.

Regarding an electric furnace, the authors mention that the behavior of arsenic is similar to that of a reverberatory furnace, whereas in a flash furnace, Outokumpu type, the amount of volatilized arsenic varies between 76 and 85%, and the sorbed arsenic varies between 7 and 17%.

The authors emphasize that the behavior of arsenic in the melting furnaces is controlled by the fact that the reactions occur in the presence of copper, since As is more stable in Cu than in $Cu_2S$, due to a decrease in its thermodynamic activity and, therefore, in its volatilization; therefore, the behavior of arsenic in melting processes varies with the amount of matte produced.

Because of the above, in the Noranda converter, arsenic removal is favorable at low matte grades. Since arsenic volatilization plays an important role in its removal during matte production in melting furnaces, any factor that increases volatilization will further enhance arsenic removal. For example, mineralization, some As compounds have higher vapor pressures than others and will therefore volatilize to a greater degree as well as the temperature of the reactor/melting furnace; and the exposure of the slag surface, the greater the exposure of the slag to the atmosphere, the greater the volatilization of As from the slag. The molten matte contains Cu, Fe, and S as its main components with up to 3% dissolved oxygen. In addition, it contains minor amounts of impurities such as As, Bi, Pb, etc., which are not removed during smelting. The distribution of As in a copper converter between gases and slag varies widely. When converting a high-grade matte, metallic copper appears early in the cycle and acts as an As collector because As, as mentioned above, is more stable and less volatile in metallic copper than in copper sulfides. Arsenic entering the copper smelting process is volatilized and removed as metal oxides in the gas streams, or with the slag. The metal oxides reported in the gases are the result of very high temperatures associated with the pyrometallurgical processes used and the inherent volatility of As and $As_2O_3$.

Finally, they conclude that the arsenic distribution is controlled by the properties of the arsenic and its environment, and that, by using two types of concentrates, characterized by their arsenic content by weight, different distributions are generated, those of high input level (As > 0.2%), where the distribution of arsenic is highly influenced by temperature and the presence of copper; and those of low level (As < 0.2%) where arsenic is not so easily released and passes to the slag.

Nakazawa, Yazawa, and Jorgensen in 1999 [30] studied the removal of sulfide arsenic from copper concentrate, this work was studied in detail in the section on As-S in neutral

atmosphere. From it, is clear that, for high oxidation degrees, volatilizations of elemental arsenic and its sulfides are gradually replaced by $As_4O_6$. In addition, the total removal of arsenic during roasting depends directly on the temperature and the degree of oxidation used; the lower the amount of arsenic present in the system, the more difficult it is to remove it using the roasting process.

According to Weisenberg et al. [22], the volatilizations of arsenic under an oxidizing atmosphere are in the elemental or sulfurized form; and, subsequently, these are oxidized in the gas phase to $As_2O_{3(g)}$; while Nakazawa et al. [30] likewise indicate that the volatilizations are in the elemental or sulfurized form; however, they present the $As_4O_{6(g)}$ phase as the predominant final phase when the oxygen concentration increases. Therefore, it can be concluded that the $As_2O_{3(g)}$, generated during the oxidation of the volatilized gas phase, forms the $As_4O_{6(g)}$ dimer, which is its stable form, as can be seen in Figure 3.

### 3.2. Arsenic Sulfides

As mentioned above, Nakazawa et al. (1999) [30] performed a simulation using HSC [47] for the removal of arsenic, bismuth, and antimony during the roasting of copper concentrates, obtaining that as oxidation degree increases, elemental arsenic volatilizations and its sulfides are gradually replaced by $As_4O_6$. For the oxidation of $As_2S_3$ under 700 °C and $PSO_2$ = 0.1 bar, arsenic is oxidized to AsS in low oxygen atmospheres; in turn, with increasing oxygen content AsS is oxidized to the $As_2O_3$ form. However, at temperatures above 700 °C AsS decomposes to As and S, which subsequently oxidizes to $As_2O_3$. At pressures of 10–5.35 bar, non-volatile $As_2O_4$ is formed. When $Fe_2O_3$ is present in the system, arsenates are formed, which are a stable, non-volatile product, according to the following reaction:

$$As_2O_{3(s)} + Fe_2O_{3(s)} + O_{2(g)} = 2FeAsO_{4(s)}$$

The vapor pressure of the volatile species As, Sb, and Bi were calculated using the HSC Chemistry program [47]; arsenic species have high vapor pressures compared with Bi and Sb volatilizations, and almost complete volatilization is expected to occur at roasting temperatures.

It is clear that Sb and Bi compounds are much less volatile than As; at low temperatures, $As_4O_{6(g)}$ has the highest vapor pressure, however, at high temperatures $As_{4(g)}$ is the predominant phase; and, $As_{2(g)}$ predominates at melting temperatures. Due to the relatively low vapor pressure of $As_2S_{3(g)}$, volatilization will proceed at a faster rate from $AsS_{(s)}$ than from $As_2S_{3(s)}$. Species such as $FeAs_2O_4$, $As_2O_4$, and $As_2O_5$ are non-volatile.

The authors emphasize that it is well known that the removal of As from copper concentrates depends on the degree of oxidation and that the optimum conditions for its removal have not yet been established. As an example, a copper concentrate with 5.3% As, which was oxidized in air at 700 °C, was simulated in the HSC Chemistry program [47]; obtaining that the As behavior was as follows; when the degree of oxidation is low, the predominant volatilized species were $As_2S_3$, $As_4S_4$, $As_4$, and $As_2$; which were gradually replaced by $As_4O_6$ as the degree of oxidation increased. When the oxygen content was set between 2400 and 3600 moles, the predominant species was $As_4O_6$. If there is enough oxygen in the system for all the $Cu_2S$ in the concentrate to be oxidized, the oxygen equilibrium pressure will be high and non-volatile Fe-$AsO_4$ will be formed as a stable species instead of $As_4O_{6(g)}$. When the amount of oxygen moles exceeded 3600, the calculations showed that all arsenic can be removed. In order to study the temperature, a 2000 moles oxygen loading was simulated between 500 and 900 °C, where it was obtained that the predominant species at low temperatures was $As_4O_6$ and then it is gradually replaced by $As_4$ and $As_2$.

Wasson et al. in 2005 [27] studied the emissions of chromium, copper, and arsenic from the open burning of Chromated Copper Arsenate (CCA) treated wood. They experimentally simulated the combustion of wood waste in order to characterize the emission composition, particle size distribution of the ash, and the resulting partitioning of arsenic, copper, and chromium between the fly ash and the residual ash. CCA was one of the main wood

preservatives in the United States from 1940 until 2003, when it was discontinued for this type of application; this solution was composed of $CrO_3$ (~47%), $CuO$ (~19%), and $As_2O_5$ (~34%), depending on the wood using a retention between 4.0 and 40.0 kg/m$^3$ of AAC—i.e., between 4.0 and 40.0 kg of preservative absorbed per cubic meter of wood; but in general most woods were treated with retention of 6.4 kg/m$^3$. This type of wood is not defined as hazardous waste, so it is often burned for disposal, although this is not legally permitted. Between 20,000 and 70,000 ppm of arsenic have been found in campfire residues; in addition, approximately 20% of emissions from CCA-treated wood correspond to arsenic volatilization, approximately 1% to chromium, and copper; and at temperatures above 800 °C, arsenic volatilization of over 80% has been found.

For the experimental development of the tests, 7 kg of wood loaded with 6.4 kg/m$^3$ of CCA was used, which was burned for 30 min at temperatures up to 1200 °C in a box of 3 m × 2.8 m × 2.1 m with an exit duct, which had a diameter of 20.3 cm and an air extractor, where samples were taken in bags for subsequent analysis by SEM, XRD, and ICP-MS. The wood was placed in a container filled with sand on an electronic balance. Arsenic, chromium, and copper corresponded to 0.1 to 0.3% of the treated mass per sample. The resulting ash consisted of 1.3% arsenic and chromium, and 60% copper. The particle size of the resulting ash was in the range of 0.1 to 1.0 μm. The volatilization of arsenic can be affected by the presence of Fe and Ca in the wood as it forms non-volatile compounds. In this case, Fe and Ca are present in amounts between 500 and 600 ppm. They refer to simulations developed using Chemsage software [51], which predict that large amounts of arsenic would volatilize below 600 °C, and Cu and Cr volatilizations are negligible below 1200 and 1500 °C, respectively. At 1200 °C the volatilized As phases include $As_2O_5$ and $As_4O_6$. They emphasize that the equilibrium predictions may be erroneous since the thermochemical information of the species may have low accuracy in the database, however, the information delivered by this software is of great help in explaining results and predicting experimental behavior. They obtain as a result of the emissions that the amount of As, Cr, and Cu released is in the range of 190–240, 8–22, 9–13 mg/kg CCA treated wood. The residual ash is composed of approximately 8.4% As, 15.9% Cr, and 9.2% Cu, where 14% of the arsenic contained was volatilized; part of the residual arsenic may be present as arsenates with chromium. Finally, they conclude that arsenic is present in the emissions as a mixture of $As_2O_5$ and $As_4O_6$, and the total $As^{3+}$/As ratio is estimated to vary between 0.8 and 0.9; copper is present as $Cu^+$ and $Cu^{2+}$, with a total $Cu^+$/Cu ratio between 0.65 and 0.7; finally, the chromium present in the volatilizations is as $Cr^{3+}$.

In 2005, Mihajlović et al. [52] studied the isothermal and non-isothermal oxidation kinetics of $As_2S_2$, with the aim of determining the kinetic parameters and oxidation mechanisms. For this purpose, they performed experimental tests of thermal analysis, they used a synthesized sample of $As_2S_2$ from Paisy Hilendarsky Plovdiv University (Bulgaria) with the following composition, 67.22% As and 32.78% S; the phases were identified by XRD. Multiple DTA-TG-DTG tests were performed to investigate the oxidation process in a non-isothermal manner. These tests were carried out on the Derivatograph 1500 thermal analysis equipment, operated under atmospheric air, with a heating rate of 20 °C/min and a maximum temperature of 1000 K. While the isothermal tests were developed using standard equipment used for this purpose, where the sample was heated in a Mars oven, in which a defined airflow was injected into the reaction zone, the exhaust gases were taken to adsorption tanks filled with hydrogen peroxide where sulfuric acid was formed. The results of the DTA-TG-DTG tests are summarized in Figure 4; the DTA curve shows the first peak in the range of 150–320 °C and a second peak between 320 and 670 °C, which are accompanied by a mass loss which is not significant until 360 °C, after which it increases significantly and substantially; this can be seen in the TG curve.

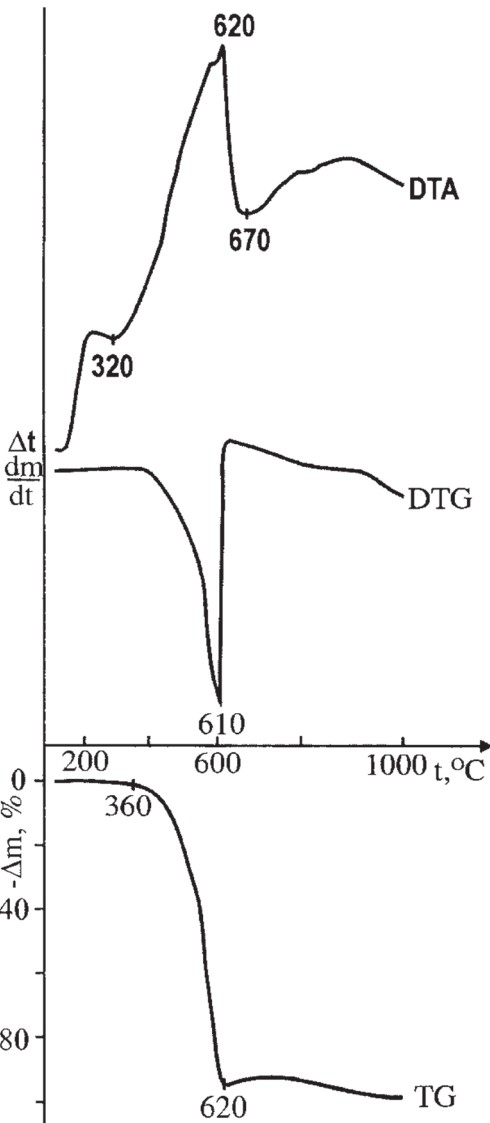

**Figure 4.** DTA-TG-DTG analysis for $As_2S_2$ [52].

According to the information provided by the literature and the figure above, it is possible to determine that the oxidation of $As_2S_2$ when heated under atmospheric air proceeds as follows:

$$2As_2S_2 + 7O_2 = 2As_2O_3 + 4SO_2$$

where above 193 °C, $As_2O_3$ forms the dimer $As_4O_{6(g)}$ according to:

$$2As_2S_2 + 7O_2 = 2As_4O_{6(g)} + 4SO_2$$

In parallel to the above reaction, up to 315 °C, the formation of $As_2O_5$ is possible according to:

$$2As_2S_2 + 9O_2 = 2As_2O_5 + 4SO_2$$

Additionally, above 321 °C, oxidation occurs according to:

$$2As_2S_{2(l)} + 7O_2 = As_4O_{6(g)} + 4SO_2$$

As mentioned, the weight loss is negligible up to 360 °C, this is because the molar mass of $As_2S_2$ is similar to that of $As_2O_3$ and $As_2O_5$. As the temperature increases, the

solubility of sulfur decreases, and according to the information in the literature at 445 °C an equilibrium in $As_4S_3$ and gaseous $SO_2$ is possible:

$$2As_2S_{2(l)} + O_{2(g)} = As_4S_{3(l)} + SO_{2(g)}$$

This is in agreement with the results obtained by the authors by ICP for a sample heated to 350 °C, which showed a composition of $As_4S_3$. Furthermore, this molten phase also reacts with oxygen forming $As_4O_6$ and $SO_2$, and above 534 °C the $As_4S_3$ begins to volatilize and oxidize, and above 650 °C only the following reaction is possible:

$$As_4S_3(g) + 6O_2 = As_4O_{6(g)} + 3SO_{2(g)}$$

According to Figure 4, the first peak between 150 and 320 °C corresponds to the formation of $As_2O_3$, $As_4O_{6(g)}$, and $As_2O_5$.

Above 360 °C, a rapid weight loss begins, which is due to the fact that $As_2O_{5(s)}$ is not stable at high temperatures, dissociating to $As_4O_{6(g)}$, and $O_{2(g)}$.

An almost linear mass loss is recorded above 534 °C up to 650 °C, which corresponds to the volatilization of $As_4S_3$.

In the second interval of the DTA curve in Figure 4, between 320 to 670 °C an exothermic peak was reported, corresponding to the formation reactions of $As_4O_{6(g)}$ from $As_2S_{2(l)}$, and $As_4S_{3(l/g)}$, together with the formation of $SO_{2(g)}$.

The slight mass increase after 620 °C can only be explained by the formation of gaseous products that must diffuse through the remaining $As_4S_3$ liquid layer into the gas and are partially trapped as bubbles. After the entire liquid phase is evaporated the mass loss in the TG curve is completed.

Therefore, it is concluded that the oxidation of $As_2S_2$ can be represented by two stages; the first is the melting of $As_2S_2$ and its transformation to $As_4S_3$, for its subsequent oxidation to $As_2O_5$ and $As_4O_{6(g)}$, and the second stage would correspond to the oxidation of $As_4S_3$ to $As_4O_{6(g)}$.

For the calculation of the kinetic parameters, they used the Borchardt and Daniels' method [53] for a non-isothermal model, and determined an activation energy of 95 kJ/mol for the temperature range between 320 and 670 °C; while for the temperature range between 350 and 450 °C, which corresponds to the transformation of $As_2S_2$ into $As_4S_3$, the activation energy was calculated isothermally obtaining 75 kJ/mol.

Later in 2009, Štrbac et al. [28] experimentally studied non-isothermally a natural mixture of $As_2S_3$ and $As_2S_2$, in order to determine the kinetics and oxidation mechanism. Initially, they emphasize that the As-S system is a subject of great interest due to the insufficient information on the oxidation mechanism reactions occurring in this system, as well as the lack of complete databases with thermodynamic and kinetic parameters describing the compounds' oxidation processes of this system. In addition, they point out that the information differs depending on the source from which they are obtained. The natural sample came from the Trepča mine, Serbia, which was characterized by XRD, obtaining that the main phases were $As_2S_3$, $As_2S_2$ together with quartz and pyrite. Non-isothermal tests were developed on Derivatograph 1500 thermal analysis equipment, with a heating rate of 20 °C/min and a maximum temperature of 1273 K. XRD analysis was performed on samples treated at 700 °C. $As_2O_3$ and $Fe_2O_3$ were detected. The DTA result obtained during the non-isothermal tests is presented in Figure 5, where two exothermic peaks connected at 276.3 and 305.6 °C, and two other peaks connected at 584.3 and 645.4 °C can be observed. These conjoined peaks represent parallel reactions, the first set corresponds to the oxidation of the two arsenic sulfides to $As_2O_3$, which is volatilized mostly as $As_4O_{6(g)}$; while the third peak corresponds to the volatilization and oxidation of the remaining arsenic sulfides to $As_4O_{6(g)}$. The fourth peak, meanwhile, represents the oxidation of pyrite to $Fe_2O_3$.

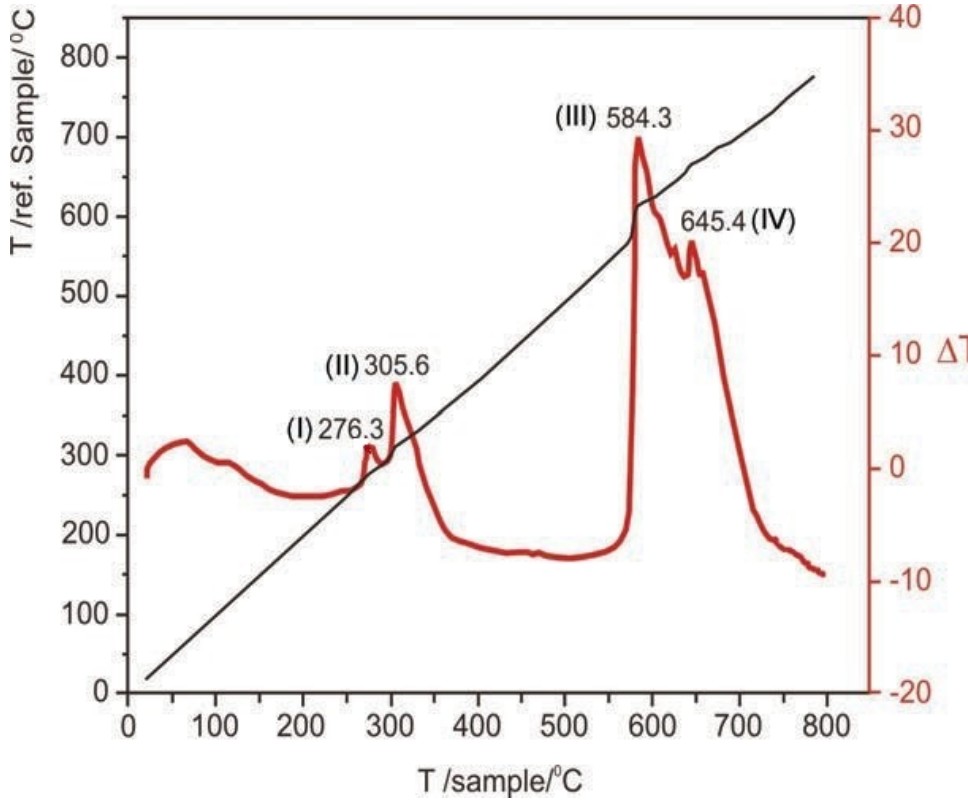

**Figure 5.** DTA results [28].

For the activation energy calculation, they use the Borchardt and Daniels' model [53] for non-isothermal systems, obtaining for the first set of reactions comprised between 260 and 330 °C, the reaction energy of 101 and 110 kJ/mol, respectively; while for the second set comprised between 580 and 745 °C, 77 and 68 kJ/mol, respectively.

From the review developed, we can corroborate what is shown in Figure 1, and what was mentioned above for the volatilized phases from elemental arsenic; the predominant volatilization is $As_4O_{6(g)}$; however, when the degree of oxidation is low, the species $As_2S_3$, $As_4S_4$, $As_4$, and $As_2$ are present, and they oxidize to $As_4O_{6(g)}$ as we increase the degree of oxidation. $As_4O_{6(g)}$ is stable at low temperatures and above 900 °C is gradually replaced by $As_4$ and $As_{2(g)}$. In addition, the possible presence of $As_2O_5$ in the gas phase is reported, but it is not stable and decomposes from its solid state into $As_2O_{3(g)}$ and $O_{2(g)}$, for the subsequent formation of the $As_4O_{6(g)}$ dimer.

*3.3. Arsenic Sulfide Compounds*

In 2001, Welhan [54] experimentally studied the mechanochemical processing of enargite to analyze the effect of grinding under different atmospheres on the solubility of enargite concentrate. Enargite is a copper ore containing arsenic. In smelting processes, ores with a high arsenic content are a problem due to the cost of removing it until it is transformed to a stable phase. Many smelters consider arsenic as a penalizing element because its content decreases the value of the concentrate, and in some cases, the value decreases to the point of not being economically viable. Generally, high levels of arsenic are linked with high antimony content, which partially substitutes arsenic, generally finding ores of the form $Cu_3(As,Sb)S_4$ and $Cu_{12}(As,Sb)_4S_{13}$. The experimental work was carried out with a commercial enargite sample, which had an approximate $P_{80}$ of 80 μm; the composition indicated by XRD was 41.5% Cu, 1.8% Fe, 28.5% S, 15.5% As, 0.76% Sb, 360 mg/kg Ag, 9.5 mg/kg Au and the remaining ~12% was composed of silicates, mainly quartz. 10.0 mg of the concentrate was loaded into a laboratory ball mill with 5 balls of 25.4 mm diameter, at room temperature (~25 °C). The atmosphere inside the mill was

completely emptied and subsequently filled with argon, air, or oxygen. The Ar and $O_2$ were treated with an overpressure to prevent air ingress from the outside. In addition, they used a modified mill with heaters, where grinding was carried out at approximately 100 °C.

They performed a computer-simulated evaluation of the thermodynamics in grinding of 10 moles of enargite during its oxidation between 25 and 100 °C, obtaining that there are two main reaction stages; at 25 °C the first stage corresponds to the oxidation to $As_2O_3$ and $CuSO_4$ and formation of elemental sulfur; while, the second stage corresponds to the oxidation of sulfur to $SO_2$, the formation of small amounts of CuS is predicted in the presence of low oxygen concentrations. The formation of $As_2O_5$ is not considered, since it was not present in the analyzed reports. Obtaining with these data the following reaction, which is highly energetic ($-9102$ kJ) and will increase the temperature of the system:

$$4Cu_3AsS_4 + 27O_2 = 12CuSO_4 + 4S + 2As_2O_3$$

They show that at 100 °C the separation of copper and arsenic occurs with less oxygen without the formation of sulfates.

$$4Cu_3AsS_4 + 7O_2 = 12CuS + 4SO_2 + 2As_2O_3$$

If this reaction were the main reaction, the arsenic could be quickly removed from the system by simple alkaline washing, or by roasting for volatilization.

According to what the authors mentioned, $SO_2$ can replace oxygen as the oxidizing agent, according to the following reaction, which is only favorable below 35 °C:

$$4Cu_3AsS_4 + 27SO_2 = 12CuSO_4 + 31S_2 + 2As_2O_3$$

The XRD of the experimentally studied enargite concentrate, without grinding, contains mainly enargite and quartz, with minor chalcopyrite and tennantite phases. Milling the concentrate for 50 h under an argon or air atmosphere at room temperature results in a slight modification of its characterization, without the formation of new phases, the peaks obtained by XRD weaken, which is typical of a granulometric redistribution. Working under an oxygen atmosphere at the same temperature, the oxidation starts only 1 h after the beginning of milling and an increase in temperature between 5 and 10 °C was observed, where $As_2O_3$ peaks are observed, which increases with increasing milling time. After 10 h CuS was found, and after 50 h $CuSO_4 \times 5H_2O$. When working the sample in an oxidizing atmosphere and at 100 °C for one hour, $As_2O_3$ and tennantite were observed as the main phases. It was concluded that the milling of enargite concentrates is of great help for the preparation of the concentrate, however, it generates high amounts of $SO_2$ emissions. Subsequently, a study on grinding and leaching with 0.5 M HCl was carried out.

After one hour of grinding at room temperature, $As_2O_3$ was observed. After 50 h of milling, the enargite and chalcopyrite completely disappeared, and oxidation to CuS, $CuSO_4$, and $As_2O_3$ was observed, of which only CuS is insoluble during leaching. After leaching for one hour at 100 °C, it can be observed that all the enargite and chalcopyrite were oxidized, and the predominant phases are tennantite, quartz, and traces of covellite. Finally, they conclude that the leaching of enargite concentrate during milling modifies its reaction mechanisms, in agreement with the reactions initially planned by the authors.

In 1983, Chakraborti et al. [25] studied the thermodynamics of the Fe-As-S-O system, focusing on the effects of particle size and temperature in order to investigate the effects of different types of atmospheres during the arsenic's decomposition and oxidation from arsenopyrite. For this purpose, they used natural and synthetic arsenopyrite; the synthetic sample was prepared in the laboratory from iron, arsenic, and sulfur powders 0.0029 inch, which had a purity of 99.99% or higher. The natural sample used came from Gold Hill, Utah; which is associated with large amounts of silica and pyrite. This sample was crushed into +1/8 inch pieces and the pyrite was manually separated. The final sample used contains 20% As comprising FeAsS, along with considerable amounts of pyrite and silica. Under

0.3 g of samples was roasted in different furnaces and atmospheres for periods of 15 to 120 min under oxidizing ($O_2$), reducing ($CO/CO_2$), and inert atmospheres (helium). The arsenic volatilizations were immediately transported and condensed for further chemical analysis. The Fe-As-S-O system involves multiple volatile species, the authors state that arsenic can be removed in the gas phase as $As_4S_4$, $As_2S_3$, $AsS$, $As_4O_6$, $As_4$, $As_2$, etc; where the nature of the emission depends on the activities of oxygen, sulfur, and arsenic during roasting. The analysis of this process is complicated due to the lack of thermodynamic information; however, it is possible to represent the equilibrium conditions by means of predominance diagrams, which can be used to evaluate the roasting process. Similar to most gas–solidus reactions, arsenic removal from arsenopyrite increases with increasing temperature. The influence of temperature on the removal kinetics is large, and for particles between 0.0029 and 0.0021 inch it is twice as fast as for particles between 0.0049 and 0.0041 inch; which is a direct consequence of the variation in surface area.

Under an inert atmosphere, arsenopyrite roasting occurs by volatilization of $As_{4(g)}$, according to the following reaction.

$$4FeAsS_{(s)} = 4FeS_{(s)} + As_{4(g)}$$

During the process, elemental arsenic is precipitated, and the possibility of the formation of gaseous arsenic sulfides is ruled out. However, an inert atmosphere does not ensure the decomposition of arsenopyrite with the formation of FeS and $As_4$, because sulfides present in the initial sample, such as FeS, can cause a high $S_{2(g)}$ pressure, which can cause arsenic to be trapped in a liquid As-S mixture, which would not allow arsenic to escape to the vapor phase.

Under a reducing atmosphere, the rate and removal of arsenic are increased compared with an inert atmosphere. By adding $CO_{(g)}$ to a neutral atmosphere, the kinetics is enhanced, however, by adding $CO_{2(g)}$ the kinetics decreases. The addition of $CO_{(g)}$ and $CO_{2(g)}$ to an inert atmosphere generates the formation of arsenic sulfide vapors and the formation of $Fe_3O_4$. Therefore, it is concluded that the slight presence of oxygen in the atmosphere volatilizes arsenic sulfides to the gas phase, such as $AsS$, $As_4S_4$, and $As_2S_3$, according to the following reactions:

$$FeAsS_{(s)} + CO_{2(g)} = FeO_{(s)} + CO_{(g)} + AsS_{(g)}$$

$$4FeAsS_{(s)} + 4CO_{2(g)} = 4FeO_{(s)} + 4CO_{(g)} + As_4S_{4(g)}$$

$$3FeAsS_{(s)} + 2CO_{2(g)} = 2FeO_{(s)} + FeAs_{(s)} + 2CO_{(g)} + As_2S_{3(g)}$$

$$FeAsS_{(s)} + 2CO_{2(g)} = FeAs_{(s)} + SO_{2(g)} + 2CO_{(g)}$$

Under oxidizing atmosphere, at high $O_{2(g)}$ pressures arsenic can be retained in the solid products by the formation of $FeAsO_4$ or $As_2O_5$. At oxygen pressures of $PO_2 = 10–21$ atm, the predominant arsenic species in the gas phase was $As_4$; whereas at $PO_2 > 10–21$ atm, the predominant arsenic species in the gas phase was $As_4O_6$.

The authors conclude that an increase in temperature increases the percentage of arsenic removal. Furthermore, inert, reducing and oxidizing atmospheres can be used for arsenic removal. In an inert atmosphere, the main volatilizations are $As_4$, $As_2$, and $S_2$. A reducing atmosphere favors the formation of arsenic sulfides in the gas phase; and an oxidizing atmosphere favors the volatilization as $As_4O_6$.

In 1988, Secco et al. [55] studied the decomposition of enargite in a neutral and oxidizing atmosphere with the objective of determining the behavior of enargite during its decomposition and oxidation. The most commonly used methods for arsenic removal from enargite concentrates are roasting and smelting where almost all the arsenic is removed in the gas phase. For the study, they used natural enargite from Saint Joe Gold Company's El Indio mine, which contained 95% enargite, 1.6% pyrite, 1.1% quartz, and 2.3% minor elements. Since both neutral ($N_2$) and oxidizing (air) atmospheres are used industrially, both were studied. The equipment used was a Setaram DTA. The samples were analyzed

by XRD because they were too small to be analyzed chemically. 2–3 g of sample were heated at a rate of 10 °C/h, the study temperatures were 500, 550, 600, and 700 °C, with an inlet gas flow of 0.1–1.0 L/min. The results obtained under a neutral atmosphere indicate that the decomposition of enargite starts at 200 °C and ends at approximately 500 °C, with an endothermic peak at 400 °C. Its decomposition occurs spontaneously at around 600 °C, according to the following reaction:

$$2Cu_3AsS_4 = 3Cu_2S + As_2S_3(g) + S_2(g)$$

For 200 min of roasting, air flows at 1 L/min, arsenic removal of 92.5, 99.0, and 99.9% is obtained at 500, 550, and 600 °C, respectively. For a flow rate of 0.1 L/min, a removal of 89.4% is obtained for 500 °C, and 97.8% for 600 °C. Arsenic retention in the calcine increases with the presence of copper and iron oxides, due to the formation of arsenates.

The results obtained for a pure oxygen atmosphere show similarity with an endothermic peak at 400 °C. According to the literature, volatilized sulfur should be extracted as $SO_2$ and arsenic volatilizes as $As_2O_3$, according to the following reaction:

$$4Cu_3AsS_4 + 13O_{2(g)} = 6Cu_2S + 2As_2O_{3(g)} + 10\,SO_{2(g)}$$

In addition, it is possible for elemental sulfur to be oxidized and arsenic to be volatilized as sulfur, according to:

$$2Cu_3AsS_4 + 2O_2 = 13Cu_2S + As_2S_{3(g)} + 2SO_{2(g)}$$

However, both reactions are exothermic and do not correspond to the recorded endothermic peak. Therefore, the same decomposition occurs as in an inert atmosphere.

Subsequently, a phase relation analysis is performed for the Cu-As-S system. In the case of the binary Cu-S system, the possible compounds were $Cu_xS$, with x = 1, 1.75–1.79, 1.97, and 2, while for the Cu-As system the compounds were $Cu_yAs$, with y = 2.375, 2.7–3.0; $Cu_4As_3$, $Cu_4As_2$, and $Cu_{8-z}As$, with 0 < z < 2.8. In the As-S system, three sulfides $As_2S_3$, AsS, and $As_4S_3$ were found. On the other hand, in the Cu-As-S system $Cu_3AsS_4$, $Cu_{12}As_4S_{13}$, $Cu_6As_4S_9$, and CuAsS were found. Reagent grade Cu, As and S were used for the experimental development of this analysis; the tests were carried out in a vacuum-sealed furnace, at temperatures between 300 to 600 °C, with a variation of 50 °C, for 2 to 35 days, and subsequently analyzed by XRD.

The authors only report the results at 600 °C, where they obtained two stable compounds tennantite and enargite, together with $Cu_yAs$, with y = 2.375, 2.7–3.0, $Cu_{8-z}As$, with 0 < z < 2.8, CuS and AsS; two liquid phases were found, one for the Cu-As system, and another for the As-S with dissolved copper sulfides, which indicates that the arsenic volatilizations coming from enargite must pass through a molten phase.

In 2007, Mihajlovic et al. [56] studied the removal of an enargite concentrate by hydrometallurgical treatment prior to roasting, intending to propose a method for arsenic removal from copper concentrates, which consists of a hydrometallurgical treatment to dissolve arsenic with sodium hypochlorite under alkaline conditions from copper concentrates prior to pyrometallurgical treatment. For this purpose, they used enargite crystals from the Bor copper mine, Serbia. It contained 26.25% Cu, 10.34% As, 19.48% S, 3.18% $Al_2O_3$, 38.12% $SiO_2$, and minor elements.

Leaching was carried out at a concentration of 0.3 mol/dm$^3$ NaClO using 800 cm$^3$ for 0.5 g of the sample between 25 and 60 °C for 120 min. Isothermal roasting tests were carried out in an electric resistance furnace with thermostatic control between 400 and 800 °C. The progress of the reaction was determined by ICP spectrometry. The solid residues were analyzed by XRD.

During pyrometallurgical tests, when the temperature in enargite roasting exceeds 193 °C, its reaction is described according to:

$$4Cu_3AsS_4 + 13O_2 = 6Cu_2S + 10SO_2 + 2As_2O_3$$

In the range 193 to 550 °C, part of the $As_2O_3$ is oxidized to $As_2O_5$, which subsequently dissociates as gases in the form of $As_4O_6$ and $O_2$, yielding the following general reaction:

$$4Cu_3AsS_4 + 13O_2 = 6Cu_2S + 10SO_2 + 2As_4O_{6(g)}$$

Finally, they propose that, for the leaching tests, the following mechanism is obtained:

$$Cu_3AsS_4 + 11NaOH + 35/2\ NaClO = 3CuO + Na_3AsO_4 + 4Na_2SO_4 + 11/2\ H_2O + 35NaCl$$

where arsenic can be stabilized by precipitating it by the following reaction.

$$Na_3AsO_4 + Ca(OH)_2 + H_2O = CaHAsO_4 + 3NaOH$$

In 2012, Padilla et al. [57] studied the mechanisms and kinetics of enargite oxidation reaction at roasting temperatures in order to understand the behavior of enargite under oxidizing atmosphere. The authors emphasize that although the thermal decomposition of enargite under inert atmosphere is well established in the literature, the literature does not present conclusive data for its decomposition under oxidizing atmosphere. The authors performed tests by measuring mass loss in a thermogravimetric analysis equipment, which consists of a vertical furnace with controlled temperature and a microbalance, in which a crucible with approximately 50 mg of sample is suspended through a quartz chain; the atmospheres used were a mixture of nitrogen and oxygen. The experimental work was carried out with pure enargite crystals from the El Indio mine, Barrick Corporation Chile; taken to a particle size distribution between −75 and +53 μm. The mineralogical composition included enargite, covellite, pyrite, and other impurities. The sample used contained 18.6% As, 46.9% Cu, and 33.0% S, which is close to the theoretical composition of the enargite 19.03% As, 48.4% Cu, and 32.5% S.

Preliminary studies were carried out for the oxidation of enargite in order to analyze its decomposition behavior under atmospheres with different oxygen concentrations. It can be observed that the behavior is similar among the weight loss curves, where each one of them is composed of three segments with different lengths and teeth: which would indicate that the oxidation of the enargite would be carried out through three stages, the first two associated with a mass loss, and the last one associated with a mass increase.

Since the first stage corresponds to a rapid weight loss, it is possible that it is affected by mass transfer. To determine this effect, the flow rate of the oxygen-nitrogen mixture injected into the system was varied from 0.6 to 1.5 L/min. The results of these tests indicated that the sample is not affected by mass transfer.

The authors subsequently studied the effect of temperature on the oxidation of enargite in the temperature range between 375 and 625 °C, in order to avoid the formation of molten phases, since the melting points of enargite and tennantite are 687 and 657 °C, respectively. The results obtained are presented in Figure 6, where it can be seen that as the temperature increases, the speed at which the three stages of decomposition occur increases. All the experiments end with a mass loss fraction of 0.45. At temperatures below 450 °C the occurrence of the second and third stages was observed, and their weight loss fraction is close to 0.4. In addition, the effect of oxygen concentration at 500 °C was studied, from which it was possible to conclude that increasing oxygen concentration causes an increase in the rate of mass loss.

To determine the reaction mechanisms and intermediate compounds, partially reacted samples were analyzed by XRD. A sample was studied at 600 °C, with a partial pressure of oxygen of 1.01 kPa and reacted for 150 and 1400 s, which would correspond to the first stage of oxidation. The sample treated for 150 s presented enargite, tennantite, and slight chalcocite peaks in its composition, whereas the sample treated for 1400 s presented tennantite and chalcocite. Another sample was studied at 600 °C, for 900 and 2700 s, under an oxygen partial pressure of 1.01 and 21.3 kPa, respectively. The 900 s sample, which

would correspond to the second oxidation stage, obtained $Cu_2S$ and $Cu_2O$, while after 2700 s, $CuO$ and $CuO \times CuSO_4$, corresponding to the third stage, were found.

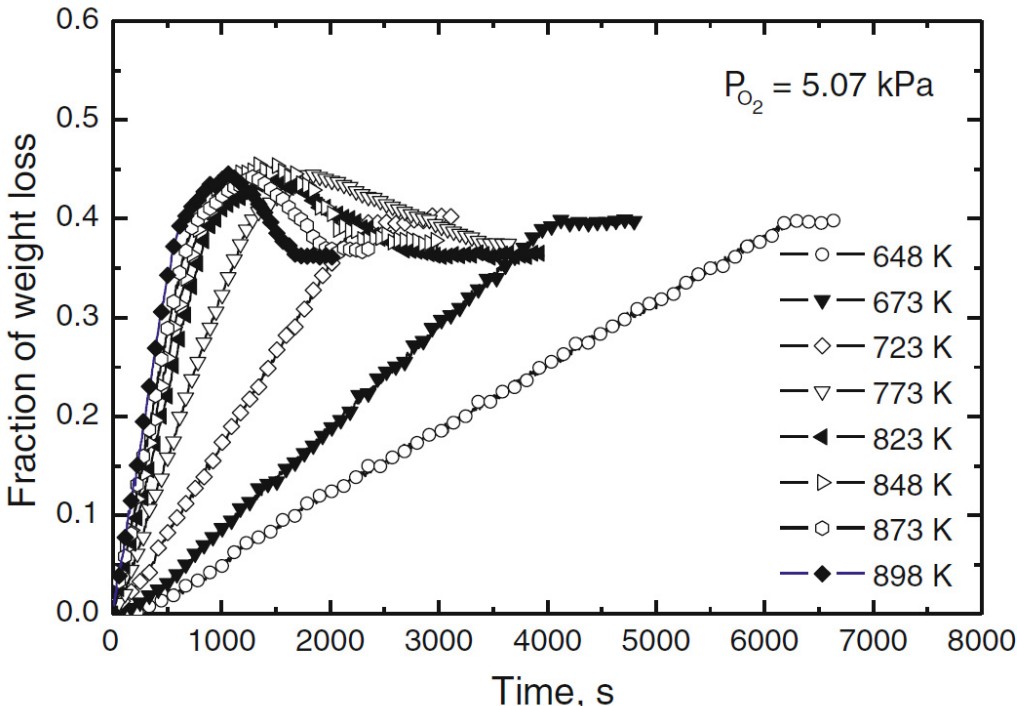

**Figure 6.** Influence of temperature on the oxidation rate of enargite [57].

These results indicate that enargite is first oxidized to tennantite as an intermediate compound, followed by the formation of chalcocite. The lack of arsenic compounds indicates that arsenic is volatilized in the first stage in the form of $As_4O_6$. In the second stage, chalcocite is oxidized to cuprite, and finally, in the third stage it is oxidized to tenorite, and traces of sulfate are formed, which would explain the mass gain recorded for this stage.

Finally, they study the reaction kinetics of the first stage, since this occurs in a linear way, therefore, it would correspond to a reaction with a constant reaction rate. For this purpose, they analyze the mass loss at different temperatures and a partial pressure of oxygen of 5.07 kPa between 375 and 625 °C, obtaining that the activation energy in the temperature range studied is 44 kJ/mol.

Finally, concluding that the thermal decomposition of enargite occurs through the following three stages.

$$4Cu_3AsS_4 + 13O_{2(g)} = 6Cu_2S + As_4O_{6(g)} + 10SO_{2(g)}$$

$$6Cu_2S + 9O_{2(g)} = 6Cu_2O + 10SO_{2(g)}$$

$$6Cu_2O + 3O_{2(g)} = 12\,CuO$$

With formation of sulfate traces.

$$6Cu_2O + 6SO_{2(g)} + 6O_{2(g)} = 6Cu_2O \bullet CuSO_{4(s)}$$

Finally, the postulated global reaction is:

$$4Cu_3AsS_4 + 25O_{2(g)} = 12CuO_{(s)} + As_4O_{6(g)} + 16SO_{2(g)}$$

For the case of the compound arsenic sulfides, we can appreciate the same trend shown in Figure 1, and previous articles regarding the predominant phase, $As_2O_{3(g)}$. Therefore,

the predominant phase for volatilizations from compound arsenic sulfides is $As_2O_{3(g)}$, in the form of its dimer $As_4O_{6(g)}$.

## 4. Discussion

### 4.1. Thermal Decomposition

As shown in Figure 1, during the thermal decomposition of elemental arsenic the predominant phase is $As_{4(g)}$, which subsequently dissociates into its allotropes, whereas for arsenic sulfides, whether compound or simple, the predominant phase is given by the $As_4S_{6(g)}$ dimer, together with $As_4S_{4(g)}$ and $As_{4(g)}$, where the predominance of each of these phases will depend on the decomposition conditions, as well as the thermal history of the sample.

In addition, based on the analyzed literature, it is proposed that one or more of the following reaction mechanisms describe the behavior of the thermal decomposition of elemental arsenic and arsenic sulfides; where the alphabetical variables refer to stoichiometric values ($1 < x < 4$; $1 < y < 5$; $1 < z < 5$; $0 < n, m < 4$).

Elemental arsenic decomposition:

$$4As_{(s)} = As_{4(g)}$$

$$As_{4(g)} = As_{2(g)}$$

$$As_{2(g)} = As_{(g)}$$

Decomposition to arsenic and elemental sulfur:

$$2As_xS_{y(s,l)} = 2As_{x(g)} + yS_{2(g)}$$

Formation of gaseous sulfides from arsenic and elemental sulfur:

$$As_{x(g)} + y/2\, S_{2(g)} = As_xS_{y(g)}$$

Decomposition of arsenic sulfides to a more stable phase releasing elemental sulfur.

$$As_xS_{y(s)} = As_xS_{(y-z)\,(s,g)} + z/2\, S_{2(g)}$$

Formation of a molten phase:

$$As_xS_{y(s)} = As_xS_{y(l)}$$

Volatilizations from molten phase:

$$As_xS_{y(l)} = As_{(x-n)}\, S_{(y-z-m)\,(l)} + As_nS_{m(g)} + (m+z)/2\, S_{2(g)}$$

Formation of $As_4S_6$ dimers.

$$2As_2S_{3(g)} = As_4S_{6(g)}$$

Decomposition of gaseous sulfides to their stable phase:

$$As_xS_{y(g)} = As_xS_{(y-z)\,(g)} + z/2\, S_{2(g)}$$

### 4.2. Oxidation

As could be observed in Figure 1, during the oxidation of elemental arsenic and its sulfides the predominant phase is $As_2O_{3(g)}$, in the form of its stable form, the dimer $As_4O_{6(g)}$.

From the review developed, we can corroborate what is shown in Figure 1, and what was previously mentioned for the volatilized and oxidized phases from elemental arsenic; the predominant volatilization is $As_4O_{6(g)}$; however, when the degree of oxidation is low,

thermal decomposition of these species occurs, volatilizing elemental and sulfide species which are subsequently oxidized to $As_2O_{3(g)}$.

For the case of the arsenic sulfide compounds, we can appreciate the same trend shown in Figure 1, and previous articles regarding the predominant phase, $As_2O_{3(g)}$. Therefore, the predominant phase for volatilizations from compound arsenic sulfides is $As_2O_{3(g)}$, in the form of its dimer $As_4O_{6(g)}$.

For the As-S-O system, it is possible to posit that the oxidation of arsenic sulfides can be described by one to several of the following reaction mechanisms. ($1 < x < 4$; $y = 3.5$; $z = 0, 2$)

Oxidation of elemental arsenic and its sulfides to the form of arsenic (III) and (V) oxide.

$$2/x \, As_{x(s)} + y/2 \, O_{2(g)} = As_2O_{y(s,g)}$$

$$As_2S_{y(s)} + y/2 \, O_{2(g)} = As_2O_{y(s,g)} + y/2 \, S_{2(g)}$$

$$1/2 \, S_{2(g)} + O_{2(g)} = SO_{2(g)}$$

Oxidation to a more stable phase

$$As_2O_{3(s)} + O_{2(g)} = As_2O_{5(s)}$$

Formation of a molten phase.

$$As_xS_{y(s)} = As_xS_{y(l)}$$

Volatilizations from molten phase.

$$As_xS_{y(l)} = As_{x(g)} + y/2 \, S_{2(g)}$$

$$As_xS_{y(l)} = As_xS_{(y-z)(g)} + z/2 \, S_{2(g)}$$

Oxidation of molten phase volatilizations.

$$As_{x(g)} + y/2 \, O_{2(g)} = As_xO_{y(g)}$$

$$As_xS_{y(g)} + yO_{2(g)} = As_2O_{y(g)} + y/2 \, SO_{2(g)}$$

$$1/2 \, S_{2(g)} + O_{2(g)} = SO_{2(g)}$$

Formation of dimers of the volatilized trioxide.

$$As_2O_{3(g)} = As_4O_{6(g)}$$

Decomposition of the oxides to their stable phase.

$$As_2O_{5(s)} = As_2O_{3(g)} + O_{2(g)}$$

## 5. Conclusions

Many works related to the roasting of compounds and the arsenic element itself were evaluated to know the volatilization of this element. Two paths involving the behavior of arsenic (as an element or compound) were obtained, summarized in the volatilization of As in a neutral atmosphere as the oxidation and subsequent volatilization of this element in an oxidizing atmosphere. It can be stated that the predominant phase during thermal decomposition (neutral atmosphere) of elemental arsenic is $As_{4(g)}$. For arsenic sulfides, $As_2S_{3(g)}$ in the form $As_4S_{6(g)}$, $As_4S_{4(g)}$, and $As_{4(g)}$.

While for an oxidizing atmosphere, depending on the oxygen concentration, the volatilization is identical to those of thermal decomposition, and subsequently oxidized in the gas phase to $As_2O_{3(g)}$, and subsequently form its dimer. For high oxygen potentials,

the predominant phase is $As_2O_{3(g)}$ in the form of $As_4O_{6(g)}$, which at high temperatures is displaced by $As_{2(g)}$ and $As_{(g)}$.

Two studies were added that made up metallic elements, arsenic and sulfur. These compounds (enargite and arsenopyrite) have relevant data showing different arsenic volatilization behaviors.

Based on the work presented, the different behaviors of arsenic in different atmospheres as well as accompanied by different metals suggest that the mineralogical analysis of raw materials should be prioritized in order to use the best roasting conditions in order to obtain volatilization and/or arsenic neutralization.

**Author Contributions:** Conceptualization, Á.A., O.J., E.B. and K.C.; methodology, Á.A., O.J., E.B. and K.C.; formal analysis, Á.A., O.J., E.B., K.C. and M.P.-T.; resources, Á.A., O.J. and E.B.; writing—original draft preparation, K.C.; writing—review and editing, Á.A., O.J., E.B., M.P.-T. and K.C.; supervision, Á.A., O.J., E.B. and K.C.; funding acquisition, Á.A., O.J. and E.B. All authors have read and agreed to the published version of the manuscript.

**Funding:** This research and the APC were funded by ANID FONDECYT INICIACION, grant number 11180432.

**Data Availability Statement:** Data presented in this study are available upon request from the corresponding author.

**Acknowledgments:** Authors thank ANID FONDECYT INICIACIÓN, 11180432 project.

**Conflicts of Interest:** The authors declare no conflict of interest.

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
