# Peer review of "Behavior of As/AsxSy in Neutral and Oxidizing Atmospheres at High Temperatures—An Overview"

_metals, doi:10.3390/met12030457_

Round 1

Reviewer 1 Report

The manuscript reviewed the main phases and reaction mechanisms during thermal decomposition and oxidation of elemental arsenic and arsenic sulfides. This is important for arsenic removal in the smelting industry, especially in copper smelting processes. This manuscript is recommended for publication in this journal with the following amendments.

(1) Many of the figures provided in the text are fuzzy, such as Figures2, 4, 5, 6

(2) A large number of arsenic-bearing copper ores contain arsenic sulfide, and the author only cites enargite may not be sufficient.

(3)The conclusions of the article are too simplistic for a review, and some outlooks should be added.

Author Response

Dear Reviewer,

Thank you very much for helping us to generate and improve this research manuscript. It is very important for us to have valuable comments from an expert in the field. Below we send our changes and/or suggestions: (in blue color):

1) Many of the figures provided in the text are fuzzy, such as Figures2, 4, 5, 6.

The referee is right. We have improved figures 2, 4, 5 and 6. Thank you very much for the comment.

2) A large number of arsenic-bearing copper ores contain arsenic sulfide, and the author only cites enargite may not be sufficient.

We have found that the work should be focused on two systems: As-S and As-S-O. However, we also intend to give two examples where two metals of interest to our region are found: Copper (Cu3AsS4) and iron (FeAsS). If we add a greater amount of sulfides that are accompanying the arsenic, we will have two problems: It will divert the main purpose of the study on arsenic (copper and iron would have to be added) and there are few concrete studies (published) on these sulfides. Thanks a lot.

3) The conclusions of the article are too simplistic for a review, and some outlooks should be added.

This comment is correct. We have modified and added several conclusions to the work.

Reviewer 2 Report

The article summaries the data regarding the reactions taking place in As-S and As-S-O systems during their oxidation and/or thermal decomposition. The overview is very detailed making the article of interest to those dealing with copper mining and metallurgy. 

There are some issues which should be addressed by Authors.

There is an error in the chemical formula for tennantite (line 79).

Figure 3 is hard to understand. The left box "oxidation" is misleading by suggesting that arsenic sulphides react with oxygen leading to ... arsenic sulphides. Please correct.

The cited literature is adequate, but in my opinion it misses some of the positions, e.g. doi: 10.1007/s12520-017-0534-1 or 10.1007/s12520-017-0534-1 (if Authors have access to them).

Language is generally correct, but there are minor language flaws which should be corrected, see eg "formation heat value" line 652 or a very long sentences in lines 674-678 or 682-686 which should be better broken into shorter ones.

Author Response

Dear reviewer, thank you very much for helping us to generate and improve this research manuscript. It is very important for us to have valuable comments from an expert in the field. Below we send our changes and/or suggestions: (in blue color)

  1. There is an error in the chemical formula for tennantite (line 79):

The referee is right. Fixed chemical formula for tennantite (line 79). Thanks for the observation

  1. Figure 3 is hard to understand. The left box "oxidation" is misleading by suggesting that arsenic sulphides react with oxygen leading to ... arsenic sulphides. Please correct.

Figure 3 was modified for better understanding. Thanks a lot.

  1. The cited literature is adequate, but in my opinion it misses some of the positions, e.g. doi: 10.1007/s12520-017-0534-1 or 10.1007/s12520-017-0534-1 (if Authors have access to them)

We have considered the comment, for this reason each DOI entered in the article has been evaluated and validated, that is, whoever needs to search for the articles will be able to do so without any inconvenience. Thanks for the help in improving our article.

  1. Language is generally correct, but there are minor language flaws which should be corrected, see eg "formation heat value" line 652 or a very long sentences in lines 674-678 or 682-686 which should be better broken into shorter ones.

Made most of the language fixes. Thanks a lot. 

Reviewer 3 Report

The manuscript metals-1574796 presents a review of As speciation and behavior during high-temperature treatment of As-bearing ores, predominantly - Cu ores. The authors, for the unknown reason, completely ignore a review of secondary (hypergene) As mineralization - natural arsenates, bearing (AsO4) anions, which are very rich and diverse especially in the copper mines of Chile. These compounds are comprised by the dozens of different arsenates, and constitute a substantial part of volatile As in the Chilean ores. The authors are invited to append a good review of this source of volatile arsenic contamination. For the introductory purposes (preliminary papers search), there is a good internet resources, e.g.:

https://www.mindat.org/search.php?search=Iquique

https://www.mindat.org/loc-657.html

For the high stability, speciation and behavior of arsenate phases under pyrometallurgical conditions, the authors may consult the following paper:

Pekov et al. (2018) Fumarolic arsenates:  a special type of arsenic mineralization. Eur. J. Mineral. 2018, 30, 305–322.

doi 10.1127/ejm/2018/0030-2718

Author Response

Dear Reviewer, thank you very much for helping us to generate and improve this research manuscript. It is very important for us to have valuable comments from an expert in the field. Below we send our changes and / or suggestions: (in blue color)

The manuscript metals-1574796 presents a review of As speciation and behavior during high-temperature treatment of As-bearing ores, predominantly - Cu ores. The authors, for the unknown reason, completely ignore a review of secondary (hypergene) As mineralization - natural arsenates, bearing (AsO4) anions, which are very rich and diverse especially in the copper mines of Chile. These compounds are comprised by the dozens of different arsenates, and constitute a substantial part of volatile As in the Chilean ores. The authors are invited to append a good review of this source of volatile arsenic contamination. For the introductory purposes (preliminary papers search), there is a good internet resources, e.g.:

https://www.mindat.org/search.php?search=Iquique

https://www.mindat.org/loc-657.html

For the high stability, speciation and behavior of arsenate phases under pyrometallurgical conditions, the authors may consult the following paper:

Pekov et al. (2018) Fumarolic arsenates:  a special type of arsenic mineralization. Eur. J. Mineral. 2018, 30, 305–322. doi 10.1127/ejm/2018/0030-2718

The main objective of our work was to present the volatilization of arsenic from different sources for different atmospheres, which considers the analysis of studies on pyrometallurgical processes on the behavior of arsenic volatilization. Including data on secondary (hypergenic) As mineralization related to volcanic fumaroles would generate a path different from the main one given in this study. However, the analysis of mineralogical data has become of interest (as shown in the work of Pekin et al., 2018), which we could include in our next work. Thank you very much for the comment delivered by the referee

Round 2

Reviewer 3 Report

Dear authors,

It seems that you even did not read my comments and related references. The secondary ARSENATES are very common and rich in Cu ores of Chilean deposits. Their thermal behavior/decomposition is completely different from the behavior of ARSENIDES from these ores. Fumarolic aresnates are herein given just as an EXAMPLE of high-temperature behavior of arsenate arsenic.

Author Response

Dear Reviewer,
The main objective of our manuscript is related to the fact that it only considers sulfurized arsenic species that, despite the existence of other species such as those mentioned by the reviewer, these sulfides mainly carry As in national copper sulfide minerals.
On the other hand, we have considered the indication of the referee (species such as arsenates) in order to continue building a work that will remain for a future article.
In order not to generate further confusion, we have decided to change only the title of the manuscript from "As/AsxOy" to "As/AsxSy", since this work was originally subdivided as: Elemental arsenic; Arsenic Sulfides; Arsenic sulfide compounds for different atmospheres (Thermal Decomposition and Oxidation).

This manuscript is a resubmission of an earlier submission. The following is a list of the peer review reports and author responses from that submission.